# On the Power of Learning-Augmented Search Trees

**Jingbang Chen** [* 1] **Xinyuan Cao** [* 2] **Alicia Stepin** [3] **Li Chen** [4]

## Abstract

We study learning-augmented binary search trees (BSTs) via Treaps with carefully designed priorities. The result is a simple search tree in which the depth of each item $x$ is determined by its predicted weight $w_x$. Specifically, each item $x$ is assigned a composite priority of $-\lfloor \log \log(1/w_x) \rfloor + U(0, 1)$ where $U(0, 1)$ is the uniform random variable. By choosing $w_x$ as the relative frequency of $x$, the resulting search trees achieve static optimality. This approach generalizes the recent learning-augmented BSTs [Lin-Luo-Woodruff ICML '22], which only work for Zipfian distributions, by extending them to arbitrary input distributions. Furthermore, we demonstrate that our method can be generalized to a B-Tree data structure using the B-Treap approach [Golovin ICALP '09]. Our search trees are also capable of leveraging localities in the access sequence through online self-reorganization, thereby achieving the working-set property. Additionally, they are robust to prediction errors and support dynamic operations, such as insertions, deletions, and prediction updates. We complement our analysis with an empirical study, demonstrating that our method outperforms prior work and classic data structures.

## 1. Introduction

The development of machine learning has sparked significant interest in its potential to enhance traditional data structures. First proposed by Kraska et al. (2018), the notion of *learned index* has gained much attention since then (Kraska et al., 2018; Ding et al., 2020; Ferragina & Vinciguerra, 2020). Algorithms with predictions have also been developed for an increasingly wide range of problems, including shortest path (Chen et al., 2022a), network flow (Po-

lak & Zub, 2024; Lavastida et al., 2021), matching (Chen et al., 2022a; Dinitz et al., 2021; Aamand et al., 2022), spanning tree (Erlebach et al., 2022), and triangles/cycles counting (Chen et al., 2022b), with the goal of obtaining algorithms that get near-optimal performances when the predictions are good, but also recover prediction-less worst-case behavior when predictions have large errors (Mitzenmacher & Vassilvitskii, 2022).

The problem of using learning to accelerate search trees, as in the original learned index question, has been widely studied in the field of data structures, focusing on developing data structures optimal to the input sequence. Mehlhorn (1975a) showed that a nearly optimal static tree can be constructed in linear time when exact key frequencies are provided. Extensive work on this topic culminated in the study of dynamic optimality. Tango trees (Demaine et al., 2007) achieve a competitive ratio of $O(\log \log n)$ while splay trees (Sleator & Tarjan, 1985) and Greedy BSTs (Lucas, 1988; Munro, 2000; Demaine et al., 2009) are conjectured to be within constant factors of optimal.

Treaps, introduced by Aragon & Seidel (1989), is a class of balanced BSTs distinguished by its use of randomization to maintain a low tree height. Each node in a Treap is assigned not only a key but also a randomly generated priority value. This design enables Treaps to satisfy the *Heap* property, ensuring that every node has a lower priority than its parent. In general, Treaps use randomness to ensure a low height instead of balancing the tree preemptively. More recently, Lin et al. (2022) introduced a learning-augmented Treap, demonstrating stronger guarantees compared to traditional Treaps. However, it relies on the strong assumption of the Zipfian distribution.

Inspired by this line of work, our research is driven by a series of critical questions.

- Whether a more general learning-augmented BST exists and achieves static optimality?

- Can such BST also obtain good guarantees under the dynamic settings?

- Are they robust to the errors caused by the prediction oracles?

---

*Equal contribution [1] University of Waterloo [2] Georgia Institute of Technology [3] Carnegie Mellon University [4] Independent; Part of the work by Jingbang Chen and Li Chen was done while at Georgia Tech. Correspondence to: Li Chen <lichenntu@gmail.com>.

*Proceedings of the 42$^{nd}$ International Conference on Machine Learning*, Vancouver, Canada. PMLR 267, 2025. Copyright 2025 by the author(s).

This paper addresses the questions affirmatively by developing new learning-augmented Treaps with carefully designed priority scores, which are applicable to *arbitrary* input distributions in both static and dynamic settings. In the static setting, we show that our learning-augmented Treaps are within a constant factor of the static optimal cost when incorporating a prediction oracle for the frequency of each item. The proposed Treaps are *robust* to predicted errors, where the additional cost induced by the inaccurate prediction grows linearly with the KL divergence between the relative frequency and its estimation. For the dynamic setting, where the trees can undergo changes after each access, we show that given a prediction oracle for the time interval until the next access, our data structure can achieve the working-set bound. This bound can be viewed as a strengthening of the static optimality bound that takes temporal locality of keys into account. Such dynamic BSTs are robust to the prediction oracle as well, where the performance degrades smoothly with the mean absolute error between the logarithm of the generated priorities and the ground truth priorities. Additionally, under the external memory model, our learning-augmented BST can be naturally extended to a B-Tree version via B-Treaps. Experimental results demonstrate that the proposed Treap outperforms both traditional data structures and other learning-augmented data structures, even when the predictions are inaccurate.

## 1.1. Overview

**Learning-Augmented Treaps via Composite Priority Functions.** The Treap is a tree-balancing mechanism initially designed around randomized priorities (Aragon & Seidel, 1989). When the priorities are assigned randomly, the resulting tree is balanced with high probability. Intuitively, this is because the root is likely to be picked among the middle elements. However, if some node is accessed very frequently (e.g. $10\%$ of the time), it's natural to assign it a larger priority. Therefore, setting the priority to be a function of access frequencies, as in Lin et al. (2022), is a natural way to obtain an algorithm more efficient on more skewed access patterns. However, when the priority is set exactly as the access frequency, some nodes would have super-logarithmic depth: if each element $i$ is accessed $i$ times, setting priority exactly as the frequencies results in a path of size $n$. The total time for processing this access sequence of size $O(n^2)$ degrades to $\Omega(n^3)$. Partly as a result of this, the analysis in Lin et al. (2022) was limited only to when frequencies are under the Zipfian distribution.

Building upon these ideas, we introduce a composite priority function, a mixture of the randomized priority function from Aragon & Seidel (1989) and the frequency-based priority function from Lin et al. (2022). This takes advantage of the balance coming from the randomness and manages to work without the strong assumption from Lin et al. (2022).

Specifically, we show in Theorem 2.4 that by setting the composite priority function to be

$$-\left\lfloor \log\log \frac{1}{w_x} \right\rfloor + U(0,1), \tag{1}$$

the expected depth of node $x$ is $O(\log(1/w_x))$. The predicted score $w_x \in (0,1)$, for instance, can be set as the relative frequency or probability of each item.

Our Treap-based scheme generalizes to B-Trees, where each node has $B$ instead of 2 children. These trees are highly important in external memory systems due to the behavior of cache performances: accessing a block of $B$ entries has a cost comparable to the cost of accessing $O(1)$ entries. By combining the B-Treaps (Golovin, 2009) with the composite priorities, we introduce a new learning-augmented B-Tree that achieves similar bounds under the *External Memory Model*. We show in Theorem 3.1 that for any weights over elements $\boldsymbol{w}$, by setting the priority to

$$-\lfloor \log_2 \log_B \frac{1}{w_x} \rfloor + U(0,1), \tag{2}$$

the expected depth of node $x$ is $O(\log_B(1/w_x))$. It is natural to see that our proposed data structures unify BSTs and B-Trees. For simplicity, we provide the results of B-Trees in the remaining content.

**Static Optimality of Learning-Augmented Search Trees.** We can construct static optimal B-Trees if we set $\boldsymbol{w}$ to be the marginal distribution of elements in the access sequence. That is, if we know the frequencies $f_1, f_2, \ldots, f_n$ of each element that appears in the access sequence, and let $m = \sum_i f_i$ to be the length of the access sequence, then we set the score $w_x = f_x/m$ in Equation (2) and the corresponding B-Tree has a total access cost that achieves the static optimality

$$\sum_{i \in [n]} f_i \log_B \frac{m}{f_i}.$$

**Dynamic Learning-Augmented Search Trees.** We also consider the dynamic setting in which we continually update the priorities of a subset of items along with the sequence access. Rather than a fixed priority for each item, we allow the priorities to change as keys get accessed. The setting has a wide range of applications in the real world. For instance, consider accessing data in a relational database. A sequence of access will likely access related items one after another. So even if the entries themselves get accessed with fixed frequencies, the distribution of the next item to be accessed can be highly dependent on the set of recently accessed items. Consider the access sequence

$$4, 2, 3, 4, 5, 2, 3, 4, 5, 2, 3, 4, 5, 1, 4$$

versus the access sequence

$$5, 2, 4, 2, 1, 4, 4, 5, \mathbf{3}, \mathbf{3}, \mathbf{3}, 4, 5, 4, 2$$

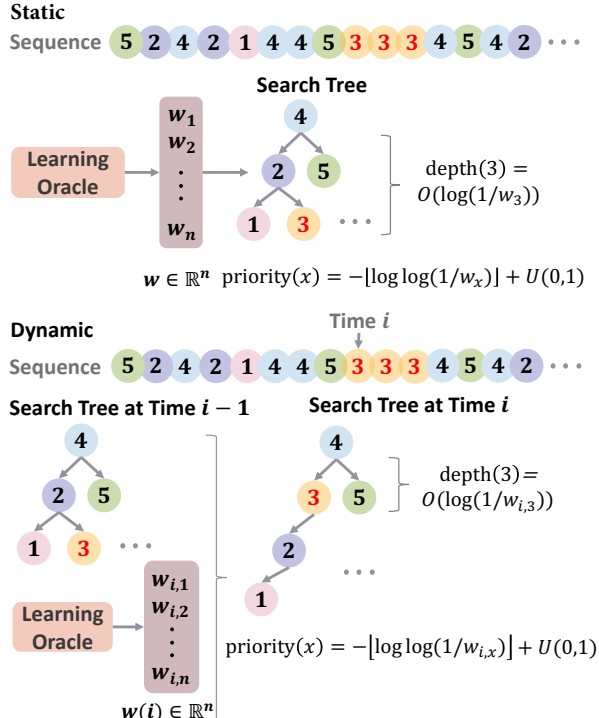

*Figure 1.* Sketch for static and dynamic learning augmented search trees. Since item 3 has a higher frequency around time $i$, dynamic search trees adjust the priority accordingly.

In both sequences, the item $4$ is accessed the most frequently. So input-dependent search trees should place $4$ near the root. However, in the second sequence, the item $3$ is accessed three consecutive times around the middle. An algorithm that's allowed to modify the tree dynamically can then modify the tree to place $3$ closer to the root during those calls. An illustration of this is in Figure 1. Note that we pay costs both when accessing the items and updating the trees. Hence, there is a trade-off between the costs of updating items' scores and the benefits of time-varying scores.

We study ways of designing composite priorities that cause this access cost to match known sequence-dependent access costs of binary trees (and their natural generalizations to B-Trees). Here, we focus on the *working-set bound*, which says that the cost of accessing an item should be, at most, the logarithm of the number of distinct items until they get accessed again. To obtain this bound, we propose a new composite priority named *working-set priority*, based on the number of distinct elements between two occurrences of the same item accessed at step $i$. We give the guarantees for the dynamic Treaps with the working-set priority in Theorem 4.4. The dynamic search Treaps further demonstrate the power of learning scores from data. While we have more data, we can quantify the dynamic environment in a more accurate way and thus improve the efficiency of the data structure.

**Robustness to Prediction Inaccuracy.** Finally, we show the robustness of our data structures with inaccuracies in prediction oracles. In the static case, we can directly relate the overhead of having inaccurate frequency predictions to the KL divergences between the true relative frequencies $p_x$ and their estimates $q_x$. This is because our composite priority can take any estimate. So plugging in the estimates $q_x$ gives the overall access cost

$$m \cdot \sum_x p_x \log_2\left(\frac{1}{q_x}\right),$$

which is exactly the cross entropy between $p$ and $q$. On the other hand, the KL divergence between $p$ and $q$ is exactly the cross entropy minus the entropy of $p$. So we get that the overhead of building the tree using noisy estimators $q$ instead of the true frequencies $p$ is exactly $m$ times the KL divergence between $p$ and $q$. We formalize the argument above in Section 2.3. We also achieve robustness results in the dynamic setting in Appendix D.2.

In all, our contributions can be summarized as follows:

- We introduce composite priorities that integrate learned advice into Treaps. The BSTs and B-Trees constructed via these priorities are within constants of the static optimal ones for arbitrary distributions (Section 2, Section 3).

- When allowing updating trees along with accessing items, we design a working-set priority function, and the corresponding Treaps with composite priorities can achieve the working-set bound (Section 4).

- Both static and dynamic learning-augmented search trees are robust to predictions (Section 2.3, Appendix D.2).

- Our experiments show favorable performance compared to prior work (Section 5).

## 1.2. Related Work

In recent years, there has been a surge of interest in integrating machine learning models into algorithm designs. A new field called *Algorithms with Predictions* (Mitzenmacher & Vassilvitskii, 2022) has garnered considerable attention, particularly in the use of machine learning models to predict input patterns to enhance performance. Examples of this approach include online graph algorithms with predictions (Azar et al., 2022), improved hashing-based methods such as Count-Min (Cormode & Muthukrishnan, 2005), and learning-augmented $k$-means clustering (Ergun et al., 2022). Practical oracles for predicting desired properties, such as predicting item frequencies in a data stream, have been demonstrated empirically (Hsu et al., 2019; Jiang et al., 2020).

Capitalizing on the existence of oracles that predict the properties of upcoming accesses, researchers are now developing more efficient learning-augmented data structures. Index structures in database management systems are one significant application of learning-augmented data structures. One key challenge in this domain is to create search algorithms and data structures that are efficient and adaptive to data whose nature changes over time. This has spurred interest in incorporating machine learning techniques to improve traditional search tree performance.

The first study on learned index structures (Kraska et al., 2018) used deep-learning models to predict the position or existence of records as an alternative to the traditional B-Tree or hash index. However, this study focused only on the static case. Subsequent research (Ferragina & Vinciguerra, 2020; Ding et al., 2020; Wu et al., 2021) introduced dynamic learned index structures with provably efficient time and space upper bounds for updates in the worst case. These structures outperformed traditional B-Trees in practice, but their theoretical guarantees were often trivial, with no clear connection between prediction quality and performance. More recently, Lin et al. (2022) proposed a learning-augmented BST via Treaps that works under the Zipfian distribution.

Other related work on BSTs analyses and B-Trees under the external memory model is included in Appendix A.

## 2. Learning-Augmented Binary Search Trees

In this section, we show that the widely taught Treap data structure can, with small modifications, achieve the static optimality conditions sought after in previous studies of learned index structures (Lin et al., 2022; Hsu et al., 2019). We start with definitions and basic properties of Treaps.

**Definition 2.1** (Treap (Aragon & Seidel, 1989)). Let $T$ be a BST over $[n]$ and priority $\in \mathbb{R}^n$ be a priority assignment on $[n]$. We say $(T, \text{priority})$ is a *Treap* if $\text{priority}_x \leq \text{priority}_y$ whenever $x$ is a descendent of $y$ in $T$.

Given a priority assignment priority, one can construct a BST $T$ such that $(T, \text{priority})$ is a Treap as follows. Take any $x^* \in \arg\max_x \text{priority}_x$ and build Treaps on $[1, x^*-1]$ and $[x^*+1, n]$ recursively using priority. Then, we just make $x^*$ the parent of both Treaps. Notice that if $\text{priority}_x$'s are distinct, the resulting Treap is unique.

*Observation* 1. Let priority $\in \mathbb{R}^n$, which assigns each item $x$ to a unique priority. There is a unique BST $T$ such that $(T, \text{priority})$ is a Treap.

From now on, we always assume that priority has distinct values. Therefore, when priority is defined from the context, the term *Treap* refers to the unique BST $T$. For each node $x \in [n]$, we use $\text{depth}(x)$ to denote its depth in $T$, i.e., the number of vertices on the path from the root to $x$.

Given any two items $x, y \in [n]$, one can determine whether $x$ is an ancestor of $y$ in a Treap without traversing the tree.

*Observation* 2. Given any $x, y \in [n]$, $x$ is an ancestor of $y$ if and only if $\text{priority}_x = \max_{z \in [x,y]} \text{priority}_z$.

Classical results from Aragon & Seidel (1989) state that if the priorities are randomly assigned, the depth of the Treap cannot be too large. Also, Treaps can be made dynamic and support operations such as insertions and deletions.

**Lemma 2.2** ((Aragon & Seidel, 1989)). *Let $U(0,1)$ be the uniform distribution over the real interval $[0,1]$. If* priority $\sim U(0,1)^n$, *each Treap node $x$ has depth $\Theta(\log_2 n)$ with high probability.*

**Lemma 2.3** ((Aragon & Seidel, 1989)). *Given a Treap $T$ and some item $x \in [n]$, $x$ can be inserted to or deleted from $T$ in $O(\text{depth}(x))$-time.*

### 2.1. Learning-Augmented Treaps

In this section, we present the construction of composite priorities and prove the following theorem.

**Theorem 2.4** (Learning-Augmented Treap via Composite Priorities). *Denote $\boldsymbol{w} = (w_1, \cdots, w_n) \in (0,1)^n$ as a score associated with each item in $[n]$ such that $\|\boldsymbol{w}\|_1 = O(1)$. Consider the following priority assignment of each item:*

$$\text{priority}_x \stackrel{\text{def}}{=} -\left\lfloor \log_2 \log_2 \frac{1}{w_x} \right\rfloor + \delta_x, \qquad (3)$$

*where $\delta_x$ is drawn independently uniformly from $(0,1)$. The expected depth of any item $x \in [n]$ is $O(\log_2(1/w_x))$.*

**Proof Plan.** Note that the priority in Equation (3) consists of two terms. We define $x$'s tier as $\tau_x := \left\lfloor \log_2 \log_2 \frac{1}{w_x} \right\rfloor = -\lfloor \text{priority}_x \rfloor$. Let $S_t = \{x \in [n] \mid \tau_x = t\}$ be the number of items whose tiers are equal $t$. We assume wlog that $\tau_x \geq 0$ for any $x$. Otherwise, $\tau_x < 0$ implies $w_x = \Omega(1)$, which can hold for only a constant number of items. So, we can always put them at the top of the Treap, which increases the depths of other items by a constant.

The expected depth of $x$ is the number of its ancestors. We show in Lemma 2.5 that the number of items at tier $t$ is bounded by $|S_t| = 2^{O(2^t)}$. Furthermore, for each tier, the ties are broken randomly due to the random offset $\delta_x \sim U(0,1)$. Then, as we show in Lemma 2.6, any item has $O(\log_2 |S_t|) = O(2^t)$ ancestors with tier $t$ in expectation. Therefore, the expected depth $\mathbb{E}[\text{depth}(x)]$ can be bound by $O(2^0 + 2^1 + \ldots + 2^{\tau_x}) = O(2^{\tau_x}) = O(\log_2(1/w_x))$.

**Lemma 2.5.** *For any integer $t \geq 0$, $|S_t| = 2^{O(2^t)}$.*

*Proof.* Observe that $x \in S_t$ if and only if

$$t \leq \log_2 \log_2(1/w_x) < t+1, \text{ and } 2^{2^t} \leq \frac{1}{w_x} < 2^{2^{t+1}}.$$

Since the total score $\|\boldsymbol{w}\|_1 = O(1)$, there are only $\text{poly}(2^{2^{t+1}}) = 2^{O(2^t)}$ such items. $\square$

Next, we bound the expected number of ancestors of item $x$ in every $S_t$ such that $t \leq \tau_x$.

**Lemma 2.6.** *Let $x \in [n]$ be any item and $t \leq \tau_x$ be a non-negative integer. The expected number of ancestors of $x$ in $S_t$ is at most $O(\log_2 |S_t|)$.*

*Proof.* First, we show that any $y \in S_t$ is an ancestor of $x$ with probability no more than $1/|S_t \cap [x, y]|$. Observation 2 says that $y$ must have the largest priority among items $[x, y]$. Thus, a necessary condition for $y$ being $x$'s ancestor is that $y$ has the largest priority among items in $S_t \cap [x, y]$. However, priorities of items in $S_t \cap [x, y]$ are i.i.d. random variables of the form $-t + U(0, 1)$. Thus, the probability that $\text{priority}_y$ is the largest among them is $1/|S_t \cap [x, y]|$.

To bound the expected number of ancestors of $x$ in $S_t$,

$$\mathbb{E}[\text{number of ancestors of } x \text{ in } S_t]$$

$$= \sum_{y \in S_t} \Pr(y \text{ is an ancestor of } x)$$

$$\leq \sum_{y \in S_t} \frac{1}{|S_t \cap [x, y]|} \leq 2 \cdot \sum_{u=1}^{|S_t|} \frac{1}{u} = O(\log_2 |S_t|),$$

where the second inequality comes from the fact that for a fixed value of $u$, there are at most two items $y \in S_t$ with $|S_t \cap [x, y]| = u$ (one with $y \leq x$, the other with $y > x$). □

Now we are ready to prove Theorem 2.4.

*Proof of Theorem 2.4.* By Lemma 2.6 and Lemma 2.5, the expected depth of $x$ can be bounded by

$$\mathbb{E}[\text{depth}(x)] = \sum_{t=0}^{\tau_x} \mathbb{E}[\text{number of ancestors of } x \text{ in } S_t]$$

$$\leq O\left(\sum_{t=0}^{\tau_x} \log_2 |S_t|\right) \leq O\left(\sum_{t=0}^{\tau_x} 2^t\right) \leq O(2^{\tau_x}).$$

We conclude the proof by observing that

$$\tau_x \leq \log_2 \log_2 \frac{1}{w_x} \leq \tau_x + 1 \text{ and } 2^{\tau_x} \leq \log_2 \frac{1}{w_x}. \quad \square$$

Moreover, our proposed learning-augmented treap supports efficient updates, where we can use rotations to do insertions, deletions, and weight changes. The following corollary follows naturally by Theorem 2.4 and Lemma 2.3.

**Corollary 2.7.** *The data structure supports insertions and deletions naturally. Suppose the score of some node $x$ changes from $w$ to $w'$ and a pointer to the node is given, the Treap can be maintained with $O(|\log_2(w'/w)|)$ rotations in expectation.*

### 2.2. Static Optimality

We present a priority assignment for constructing statically optimal Treaps given item frequencies. Given any access sequence $X = (x(1), \ldots, x(m))$, we define $f_x$ for any item $x$, to be its *frequency* in $X$, i.e. $f_x := |\{i \in [m] \mid x(i) = x\}|, x \in [n]$. For simplicity, we assume that every item is accessed at least once, i.e., $f_x \geq 1, x \in [n]$. We prove the following result as a simple application of Theorem 2.4 by setting $w_x := \frac{f_x}{m}$, $x \in [n]$.

**Theorem 2.8.** *For any item $x \in [n]$, we set its priority as*

$$\text{priority}_x := -\left\lfloor \log_2 \log_2 \frac{m}{f_x} \right\rfloor + \delta_x, \delta_x \sim U(0, 1).$$

*In the corresponding Treap, each node $x$ has expected depth $O(\log_2(m/f_x))$. Therefore, the total time for processing the access sequence is $O(\sum_x f_x \log_2(m/f_x))$, which matches the performance of the optimal static BSTs up to a constant factor.*

### 2.3. Robustness Guarantees

In practice, one could only estimate $q_x \approx p_x = f_x/m, x \in [n]$. A natural question arises: how does the estimation error affect the performance? In this section, we analyze the drawbacks in performance given the estimation errors. As a result, we will show that our Learning-Augmented Treaps are robust against noise and errors.

For each item $x \in [n]$, define $p_x = f_x/m$ to be the relative frequency of item $x$. One can view $p$ as a probability distribution over $[n]$ such that $p(x) = p_x$. Then we can restate the expected depth of each item in Theorem 2.8 using the notion of entropy. We define the entropy as follows and state the corollary in Corollary 2.10.

**Definition 2.9** (Entropy). Given a probability distribution $p$ over $[n]$, define its *Entropy* as $\text{Ent}(p) := \sum_x p_x \log_2(1/p_x) = \mathbb{E}_{x \sim p}[\log_2(1/p_x)]$.

**Corollary 2.10.** *In Theorem 2.8, the expected depth of each item $x$ is $O(\log_2(1/p_x))$ and the expected total cost is $O(m \cdot \text{Ent}(p))$, where $\text{Ent}(p) = \sum_x p_x \log_2(1/p_x)$ measures the entropy of the distribution $p$.*

Now we consider the case when we cannot access the relative frequency $p_x$. Instead, we are given $p_x$'s estimator, $q_x$, and construct the data-augmented BST with $q_x$. Similarly, we view $q$ as a data distribution over $[n]$ such that $q(x) = q_x$. Then we show that the total access of the treap built with $q_x$ equals the total access number $m$ times the cross entropy of $p$ and $q$ in Theorem 2.13. We start with some definitions.

**Definition 2.11** (Cross Entropy). Given two distributions $p, q$ over $[n]$, define its *Cross Entropy* as $\text{Ent}(p, q) := \sum_x p_x \log_2(1/q_x) = \mathbb{E}_{x \sim p}[\log_2(1/q_x)]$.

**Definition 2.12** (KL Divergence). Given two distributions $p, q$ over $[n]$, define its *KL Divergence* as $D_{\text{KL}}(p, q) = \text{Ent}(p, q) - \text{Ent}(p) = \sum_x p_x \log_2(p_x/q_x)$.

We analyze the run time given frequency estimations $q$.

**Theorem 2.13.** *Given a distribution $q$, an estimate of the true relative frequencies distribution $p$. For any item $x \in [n]$, we draw a random number $\delta_x \sim U(0,1)$ and set its priority as*

$$\mathsf{priority}_x := -\left\lfloor \log_2 \log_2 \frac{1}{q_x} \right\rfloor + \delta_x.$$

*In the corresponding Treap, each node $x$ has expected depth $O(\log_2(1/q_x))$. Therefore, the total time for processing the access sequence is $O(m \cdot \mathrm{Ent}(p, q))$.*

*Proof.* Define the weights $w_x = q_x$ for each item $x \in [n]$. Clearly, $\|w\|_1 = 1$ and we can apply Theorem 2.4 to prove the bound on the expected depths. The total time for processing the access sequence is, by definition,

$$O\left( \sum_{x \in [n]} f_x \log_2 \frac{1}{q_x} \right) = O\left( m \cdot \sum_{x \in [n]} p_x \log_2 \frac{1}{q_x} \right)$$
$$= O\left( m \cdot \mathrm{Ent}(p, q) \right). \qquad \square$$

## 2.4. Analysis of Other Priority Assignments

In this section, we discuss two different priority assignments. For each assignment, we design an input distribution that results in a greater expected depth than the expected depth with our priority assignment stated in Theorem 2.8. We define the distribution $p$ as $p(x) = p_x = f_x/m, x \in [n]$. We use $f \gtrsim g$ to indicate that $f$ is greater or equal to $g$ up to a constant factor.

The first priority assignment is used in Lin et al. (2022). They assign priorities according to $p_x$ entirely, i.e., $\mathsf{priority}_x = p_x, x \in [n]$. Assuming that items are ordered randomly, and $p$ is a Zipfian distribution, they show *Static Optimality*. However, it does not generally hold–there exists a distribution $p$ where the expected access cost for (Lin et al., 2022) is $\Omega(n)$, while our data structure (Theorem 2.4) achieves only a $O(\log_2 n)$ cost.

**Theorem 2.14.** *Consider the priority assignment that assigns the priority of each item to be $\mathsf{priority}_x := p_x, x \in [n]$. There is a distribution $p$ over $[n]$ such that the expected access time, $\mathbb{E}_{x \sim p}[\mathsf{depth}(x)] = \Omega(n)$.*

*Proof.* We define for each item $x$, $p_x := \frac{2(n-x+1)}{n(n+1)}$. One could easily verify that $p$ is a distribution over $[n]$. In addition, the smaller the item $x$, the larger the priority $\mathsf{priority}_x$. Thus, by the definition of Treaps, item $x$ has depth $x$. The expected access time of $x$ sampled from $p$ can be lower bounded as follows:

$$\mathbb{E}_{x \sim p}[\mathsf{depth}(x)] = \sum_{x \in [n]} p_x \cdot \mathsf{depth}(x)$$
$$= \sum_{x \in [n]} \frac{2(n-x+1)}{n(n+1)} \cdot x = \frac{2}{n(n+1)} \sum_{x \in [n]} x(n-x+1)$$
$$\gtrsim \frac{2}{n(n+1)} \cdot n^3 \gtrsim n. \qquad \square$$

Next, we consider the priority assignment $\mathsf{priority}_x := -\lfloor \log_2 1/p_x \rfloor + \delta_x, \delta_x \sim U(0,1)$.

**Theorem 2.15.** *Consider the following priority assignment that sets the priority of each node $x$ as $\mathsf{priority}_x := -\lfloor \log_2 1/p_x \rfloor + \delta_x, \delta_x \sim U(0,1)$. There is a distribution $p$ over $[n]$ such that the expected access time, $\mathbb{E}_{x \sim p}[\mathsf{depth}(x)] = \Omega(\log_2^2 n)$.*

*Proof.* We assume WLOG that $n$ is an even power of 2. Define $K = \frac{1}{2} \log_2 n$. We partition $[n]$ into $K+1$ segments $S_1, \ldots, S_K, S_{K+1} \subseteq [n]$. For $i = 1, 2, \ldots, K$, we add $2^{1-i} \cdot n/K$ elements to $S_i$. Thus, $S_1$ has $n/K$ elements, $S_2$ has $n/2K$, and $S_K$ has $\sqrt{n}/K$ elements. The rest are moved to $S_{K+1}$.

Now, we can define the distribution $p$. Elements in $S_{K+1}$ have zero-mass. For $i = 1, 2, \ldots, K$, elements in $S_i$ has probability mass $2^{i-1}/n$. One can directly verify that $p$ is indeed a probability distribution over $[n]$.

In the Treap with the given priority assignment, $S_i$ forms a subtree of expected height $\Omega(\log_2 n)$ since $|S_i| \geq n^{1/3}$ for any $i = 1, 2, \ldots, K$ (Lemma 2.2). In addition, every element of $S_i$ passes through $S_{i+1}, S_{i+2}, \ldots, S_K$ on its way to the root since they have strictly larger priorities. Therefore, the expected depth of element $x \in S_i$ is $\Omega((K - i) \log_2 n)$. One can lower bound the expected access time (which is the expected depth) as:

$$\mathbb{E}_{x \sim p}[\mathsf{depth}(x)] \gtrsim \sum_{i=1}^{K} \sum_{x \in S_i} p_x \cdot (K - i) \cdot \log_2 n$$
$$= \sum_{i=1}^{K} p(S_i) \cdot (K-i) \cdot \log_2 n = \sum_{i=1}^{K} \frac{1}{K} \cdot (K-i) \cdot \log_2 n$$
$$\gtrsim K \log_2 n \gtrsim \log_2^2 n,$$

where we use $p(S_i) = |S_i| \cdot 2^{i-1}/n = 1/K$ and $K = \Theta(\log_2 n)$. That is, the expected access time is at least $\Omega(\log_2^2 n)$. $\qquad \square$

# 3. Learning-Augmented B-Trees

We now extend the ideas above, specifically the composite priority notions, to B-Trees in the *External Memory Model*. The main results are shown as follows. Full details are included in Appendix C. We show that the learning-augmented B-Treaps (Appendix C.1) obtain static optimality (Appendix C.2) and is robust to the noisy predicted scores (Appendix C.3).

**Theorem 3.1** (Learning-Augmented B-Treap via Composite Priorities). *Denote $w = (w_1, \cdots, w_n) \in (0,1)^n$ as a score associated with each element of $[n]$ such that $\|w\|_1 = O(1)$ and a branching factor $B = \Omega(\ln^{1/(1-\alpha)} n)$, consider the following priority assignment scheme:*

$$\mathsf{priority}_x := -\lfloor \log_2 \log_B \frac{1}{w_x} \rfloor + \delta_x, \ \delta_x \sim U(0,1).$$

*There is a randomized data structure that maintains a B-Tree $T^B$ over $U$ such that*

1. *Each item $x$ has expected depth $O(\frac{1}{\alpha} \log_B(1/w_x))$.*

2. *Insertion or deletion of item $x$ into/from $T$ touches $O(\frac{1}{\alpha} \log_B(1/w_x))$ nodes in $T^B$ in expectation.*

3. *Updating the weight of item $x$ from $w$ to $w'$ touches $O(\frac{1}{\alpha}|\log_B(w'/w)|)$ nodes in $T^B$ in expectation.*

*In addition, if $B = O(n^{1/2-\delta})$ for some $\delta > 0$, all above performance guarantees hold with high probability $1 - \delta$. If we are given the frequency $f_x$ for each $x$ and set the priority as*

$$\mathsf{priority}_x := -\left\lfloor \log_2 \log_B \frac{m}{f_x} \right\rfloor + \delta_x, \delta_x \sim U(0,1).$$

*Then the total access cost $O(\sum_x f_x \log_B(m/f_x))$ achieves the static optimality.*

## 4. Dynamic Learning-Augmented Search Trees

In this section, we investigate the properties of dynamic search trees that permit modifications concurrent with sequence access. Prioritizing items that are anticipated to be accessed in the near future to reside at lower depths within the tree can significantly reduce access times. Nonetheless, updating the B-trees introduces additional costs. The overarching goal is to minimize the composite cost, which includes both the access operations across the entire sequence and the modifications to the B-trees. In the main content, we specifically concentrate on the study of locally dynamic B-trees, which are characterized by the restriction that tree modifications are limited solely to the adjustment of priorities for the items being accessed.

Here, we construct a dynamic learning-augmented B-trees that achieves the *working-set property*. In data structures, the working set is the collection of data that a program uses frequently over a given period. This concept is important because it helps us understand how a program interacts with memory and thus enables us to design more efficient data structures and algorithms. For example, if a program is sorting a list, the working set might be the elements of the list it is comparing and swapping right now. The size of the working set can affect how fast the program runs. A smaller working set can make the program run faster because it means the program doesn't need to reach out to slower parts of memory as often. In other words, if we know which parts of a data structure are used most, we can organize the data or even the memory in a way that makes accessing these parts faster, which can speed up the entire program.

We define the *working-set size* as the number of distinct items accessed between two consecutive accesses. Correspondingly, we design a time-varying score, *working-set*

*score*, as the reciprocal of the square of one plus working-set size. We will show that the working-set score is locally changed and there exists a data structure that achieves the *working-set property*, which states that the time to access an element is a logarithm of its working-set size.

The formal definitions and the main theorems in this section are presented as follows. We include more general results for dynamic B-trees and omitted proofs in Appendix D.

**Definition 4.1** (Previous and Next Access $\mathsf{prev}(i,x)$ and $\mathsf{next}(i,x)$)**.** Let $\mathsf{prev}(i,x)$ be the previous access of item $x$ at or before time $i$, i.e, $\mathsf{prev}(i,x) := \max\{i' \leq i \mid x(i') = x\}$. Let $\mathsf{next}(i,x)$ to be the next access of item $x$ after time $i$, i.e, $\mathsf{next}(i,x) := \min\{i' > i \mid x(i') = x\}$.

**Definition 4.2** (Working-set Size $\mathsf{work}(i,x)$)**.** Define the *working-set size* $\mathsf{work}(i,x)$ as the number of distinct items accessed between the previous access of item $x$ at or before time $i$ and the next access of item $x$ after time $i$. That is,

$$\mathsf{work}(i,x) \overset{\text{def}}{=} |\{x(\mathsf{prev}(i,x)+1), \cdots, x(\mathsf{next}(i,x))\}|.$$

If $x$ does not appear after time $i$, we define $\mathsf{work}(i,x) := n$.

Note that this definition captures the temporal locality and diversity of user behavior around item. Although it requires knowledge of future access items, it aligns conceptually with practices in recommendation systems, where temporal context is used to model user intent and predict relevance. For example, session-based recommendation often considers the diversity of items within a user's recent session (Wang et al., 2021). In practice, we can approximate this score using causal proxies (e.g., past-only working-set size) or apply machine learning models to predict it based on observable access patterns.

**Definition 4.3** (Working-set Score $\omega(i,x)$)**.** Define the time-varying score as the reciprocal of the square of one plus working-set size. That is,

$$\omega(i,x) = \frac{1}{(1 + \mathsf{work}(i,x))^2}$$

**Theorem 4.4** (Dynamic Search Tree with Working-set Priority)**.** *With the working-set size $\mathsf{work}(i,x)$ known and the branching factor $B = \Omega(\ln^{1.1} n)$, there is a randomized data structure that maintains a B-tree $T^B$ over $[n]$ with the priorities assigned as*

$$\mathsf{priority}(i,x) = -\lfloor \log_2 \log_B (1 + \mathsf{work}(i,x))^2 \rfloor + U(0,1).$$

*Upon accessing the item $x$ at time $i$, the expected depth of item $x$ is $O(\log_2(1 + \mathsf{work}(i,x))$. The expected total cost for processing the whole access sequence $\boldsymbol{X}$ is the following. The data structure satisfies the working-set property.*

$$\mathsf{cost}(\boldsymbol{X}, \omega) = O\left(n \log_B n + \sum_{i=1}^m \log_B(1 + \mathsf{work}(i,x))\right).$$

*In particular, if $B = O\left(n^{1/2-\delta}\right)$ for some $\delta > 0$, the guarantees hold with probability $1 - \delta$.*

**Remark.** Consider two sequences with length $m$, $\boldsymbol{X}_1 = (1, 2, \cdots, n, 1, 2, \cdots, n, \cdots, 1, 2, \cdots, n)$, $\boldsymbol{X}_2 = (1, 1, \cdots, 1, 2, 2, \cdots, 2, \cdots, n, n, \cdots, n)$. Two sequences have the same total cost if we have a fixed score. However, $X_2$ should have less cost because of its repeated pattern. Given the frequency freq as a time-invariant priority, by Theorem 3.1, the optimal static costs are

$$\mathsf{cost}(\boldsymbol{X}_1, \mathsf{freq}) = \mathsf{cost}(\boldsymbol{X}_2, \mathsf{freq}) = O(m \log_2 n).$$

But for the dynamic BSTs, with the working-set score, we calculate both costs from Theorem 4.4 as

$$\mathsf{cost}(\boldsymbol{X}_1, \omega) = O(m \log_2(n+1)),$$
$$\mathsf{cost}(\boldsymbol{X}_2, \omega) = O(n \log_2 n + m \log_2 3).$$

This means that our proposed priority can better capture the timing pattern of the sequence and thus perform better than the optimal static setting.

Finally, we use the following theorem to show the robustness of the results when the scores are inaccurate.

**Theorem 4.5** (Dynamic Search Tree with Working-set Priority). *Given the predicted locally changed working-set score $\widetilde{\omega}(i) \in (0, 1)^n$ satisfying $\|\widetilde{\omega}(i)\|_1 = O(1)$, $\widetilde{\omega}_{i,j} \geq 1/\mathrm{poly}(n)$ and the branching factor $B = \Omega(\ln^{1.1} n)$, there is a randomized data structure that maintains a B-Tree over the $n$ keys such that the expected total cost for processing the whole access sequence $\boldsymbol{X}$, $\mathsf{cost}(\boldsymbol{X}, \widetilde{\omega})$, is*

$$\mathsf{cost}(\boldsymbol{X}, \omega) + O\left(\sum_{i=1}^{m} \left|\log_B \omega_{i,x(i)} - \log_B \widetilde{\omega}_{i,x(i)}\right|\right).$$

*In particular, if $B = O(n^{1/2-\delta})$ for some $\delta > 0$, the guarantees hold with probability $1 - \delta$.*

## 5. Experiments

In this section, we give experimental results that compare our learning-augmented Treap (*learn-bst*) with learning-augmented BST (Lin et al., 2022) (*learn-llw*), learning-augmented skip-list (Fu et al., 2025) (*learn-skiplist*) and classical search tree data structures including Red-Black Trees (*red-black*), AVL Trees (*avl*), Splay Trees (*splay*), B-Trees of order 3 (*b-tree*), and randomized Treaps (*random-treap*). Experiments are conducted in a similar manner in Lin et al. (2022): **(1)** All keys are inserted in a random order to avoid insertion order sensitivity. **(2)** The total access cost is measured by the total number of comparisons needed while accessing keys.

We consider a synthetic data setting, with $n$ unique items appearing in a sequence of length $m$. We define the frequency

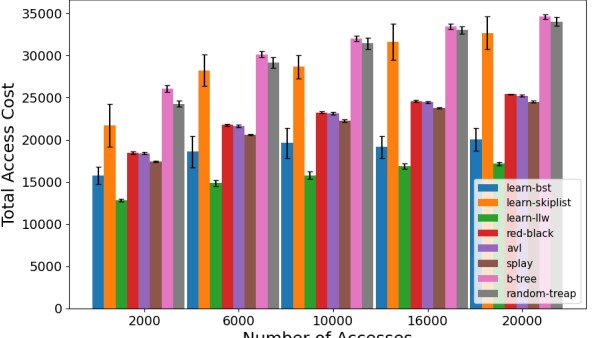

*Figure 2.* Zipfian distribution, $\alpha = 1$.

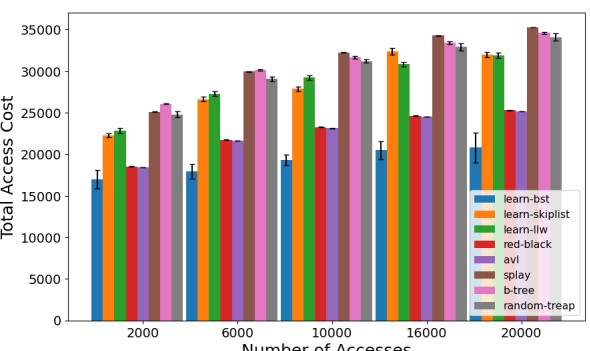

*Figure 3.* Adversarial distribution.

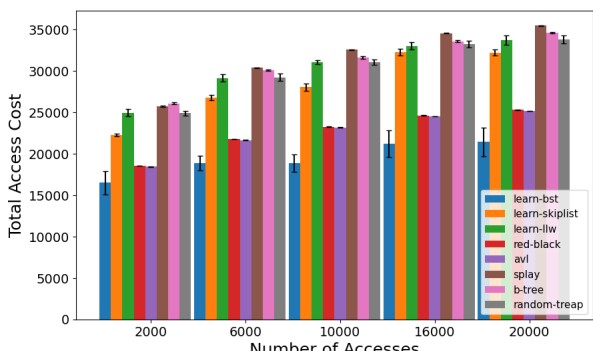

*Figure 4.* Uniform distribution.

of each item $i$ as $f_i$ and its relative frequency as $p_i = f_i/m$. All results are based on ten independent trials.

### 5.1. Perfect Prediction Oracle on Frequency

We first assume that we are given a perfect prediction oracle on the item frequency. The data follows one of three distributions: the Zipfian distribution, the distribution described in Theorem 2.14 (adversarial distribution), and the uniform distribution. We set $n = 1000$ and vary $m$ over $[2000, 6000, 10000, 16000, 20000]$. The $x$-axis represents the number of unique items, and the $y$-axis denotes the number of comparisons made, which measures access cost.

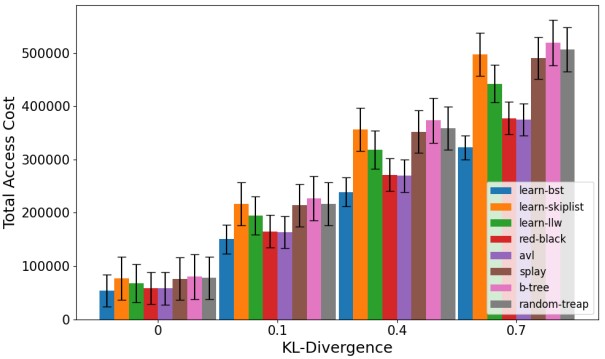

*Figure 5.* Inaccurate Prediction Oracle.

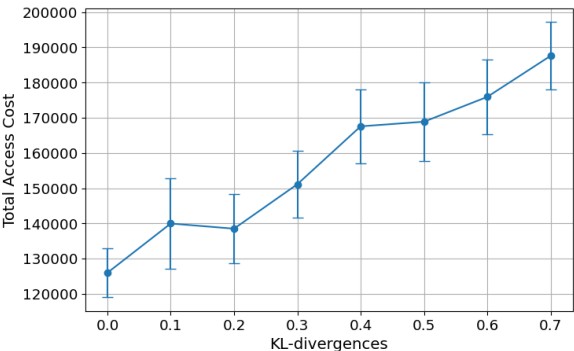

*Figure 6.* Relationship between total cost and KL divergence between true and predicted frequency using our learning-augmented treaps.

**Zipfian Distribution.** The Zipfian distribution with parameter $\alpha$ has relative frequencies $p_i = \frac{1}{i^\alpha H_{n,\alpha}}$, where $H_{n,\alpha} = \sum_{i=1}^{n} \frac{1}{i^\alpha}$ is the $n^{th}$ generalized harmonic number of order $\alpha$. In our experiment, we set $\alpha = 1$. As shown in Figure 2, our Treaps outperform all other data structures except (Lin et al., 2022), which explicitly assumes a Zipfian distribution.

**Adversarial Distribution.** In the proof of Theorem 2.14, we construct a distribution with relative frequency given by $p_i = \frac{2(n-i+1)}{n(n+1)}$. We prove that using the priority assignment as in (Lin et al., 2022), the expected depth is $\Omega(n)$. As shown in Figure 3, the data structure in Lin et al. (2022) performs significantly worse under this adversarial distribution compared to the Zipfian case, while our Treaps maintain the best performance among all data structures.

**Uniform Distribution.** We also consider uniform distribution, where each item has a relative frequency of $p_i = \frac{1}{n}$. As shown in Figure 4, our Treaps outperform all other data structures as well.

### 5.2. Inaccurate Prediction Oracle on Frequency

We consider the scenario where our learning-augmented Treaps are constructed based on an inaccurate prediction of item frequencies. The inaccuracy of the prediction is quantified using the KL divergence between the true relative frequency and the predicted relative frequency. Specifically, we initialize a uniform distribution and optimize it toward a target distribution with a specified KL divergence level using the Sequential Least Squares Programming (SLSQP) optimizer in SciPy (Virtanen et al., 2020). The $x$-axis represents the KL divergence, while the $y$-axis denotes the total access cost. We set $n = 5000$ and $m = 10000$.

As shown in Figure 5, our Treaps not only outperform the investigated alternatives but also exhibit a graceful degradation as the difference between the predicted and actual distribution increases. Additionally, we conducted other robustness experiments on mixtures of distributions, as detailed in Appendix E.

### 5.3. Total Cost and KL Divergence

In Theorem 2.13, we show that in our learning-augmented treaps, when the frequency predictor is inaccurate, the additional cost increases linearly with the KL divergence between the true and predicted frequencies. We complement this theoretical result with experiments using the same setup as in Section 5.2. Specifically, we set $n = 5000, m = 10000$ and set the KL divergence to vary over $[0, 0.1, 0.2, 0.3, 0.4, 0.5, 0.6, 0.7]$. As shown in Figure 6, the experimental results confirm that the additional cost grows linearly with the KL divergence.

## Acknowledgments

We thank Chris Lambert, Richard Peng, Mars Xiang, and Daniel Sleator for their helpful discussions and insights, and Tian Luo, Samson Zhou, and Chunkai Fu for their insights on the experimental code.

## Impact Statement

This paper presents work whose goal is to advance the field of Machine Learning. There are many potential societal consequences of our work, none of which we feel must be specifically highlighted here.

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

# A. Other Related Works

**Beyond Worst-Case Analyses of Binary Trees.** Binary trees are among the most ubiquitous pointer-based data structures. While schemes without re-balancing do obtain $O(\log_2 n)$ time bounds in the average case, their behavior degenerates to $\Omega(n)$ on natural access sequences such as $1, 2, 3, \ldots, n$. To remedy this, many tree balancing schemes with $O(\log_2 n)$ time worst-case guarantees have been proposed (Adelson-Velskii & Landis, 1963; Guibas & Sedgewick, 1978; Cormen et al., 2009).

Creating binary trees optimal for their inputs has been studied since the 1970s. Given access frequencies, the static tree of optimal cost can be computed using dynamic programs or clever greedies (Hu & Tucker, 1970; Mehlhorn, 1975b; Yao, 1982; Karpinski et al., 1996). However, the cost of such computations often exceeds the cost of invoking the tree. Therefore, a common goal is to obtain a tree whose cost is within a constant factor of the entropy of the data, multiple schemes do achieve this either on worst-case data (Mehlhorn, 1975b), or when the input follows certain distributions (Allen & Munro, 1978).

A major disadvantage of static trees is that their cost on any permutation needs to be $\Omega(n \log_2 n)$. On the other hand, for the access sequence $1, 2, 3, \ldots, n$, repeatedly bringing the next accessed element to the root gives a lower cost $O(n)$. This prompted Allen and Munro to propose the notion of self-organizing binary search trees. This scheme was extended to splay trees by (Sleator & Tarjan, 1985). Splay trees have been shown to obtain many improved cost bounds based on temporal and spatial locality (Sleator & Tarjan, 1985; Cole et al., 2000; Cole, 2000; Iacono, 2005). In fact, they have been conjectured to have access costs with a constant factor of optimal on any access sequence (Iacono, 2013). Much progress has been made towards showing this over the past two decades (Demaine et al., 2009; Derryberry & Sleator, 2009; Chalermsook et al., 2020; Bose et al., 2020).

From the perspective of designing learning-augmented data structures, the dynamic optimality conjecture almost goes contrary to the idea of incorporating predictors. It can be viewed as saying that learned advice do not offer gains beyond constant factors, at least in the binary search tree setting. Nonetheless, the notion of access sequence, as well as access-sequence-dependent bounds, provides useful starting points for developing prediction-dependent search trees in online settings. In this paper, we choose to focus on bounds based on temporal locality, specifically, the working-set bound. This is for two reasons: the spatial locality of an element's next access is significantly harder to describe compared to the time until the next access; and the current literature on spatial locality-based bounds, such as dynamic finger tends to be much more involved (Cole et al., 2000; Cole, 2000). We believe an interesting direction for extending our composite scores is to obtain analogs of the unified bound (Iacono, 2001; Bădoiu et al., 2007) for B-Trees.

**B-Trees and External Memory Model.** Parameterized B-Trees (Brodal & Fagerberg, 2003) have been studied to balance the runtime of read versus write operations, and several bounds have been shown with regard to the blocks of memory needed to be used during an operation. The optimality is discussed in both static and dynamic settings. Rosenberg & Snyder (1981) compared the B-Tree with the minimum number of nodes (denoted as *compact*) with non-compact B-Trees and with time-optimal B-Trees. Bender et al. (2016) considers keys have different sizes and gives a cache-oblivious static atomic-key B-Tree achieving the same asymptotic performance as the static B-Tree. When it comes to the dynamic setting, the trade-off between the cost of updates and accesses is widely studied (O'Neil et al., 1996; Jagadish et al., 1997; Jermaine et al., 1999; Buchsbaum et al., 2000; Yi, 2012). Bose et al. (2008) studied the dynamic optimality of B-Trees and presented a self-adjusting B-Tree data structure that is optimal up to a constant factor when $B$ is constant.

B-Treap were introduced by Golovin (2008; 2009) as a way to give an efficient history-independent search tree in the external memory model. These studies revolved around obtaining $O(\log_B n)$ worst-case costs that naturally generalize Treaps. Specifically, for sufficiently small $B$ (as compared to $n$), Golovin showed a worst-case depth of $O(\frac{1}{\alpha} \log_B n)$ with high probability, where $B = \Omega(\ln^{1/(1-\alpha)} n)$. The running time of this structure has recently been improved by Safavi & Seybold (2023) via a two-layer design.

The large node sizes of B-Trees interact naturally with the external memory model, where memory is accessed in blocks of size $B$ (Brodal & Fagerberg, 2003; Vitter, 2001). The external memory model itself is widely used in data storage and retrieval (Margaritis & Anastasiadis, 2013), and has also been studied in conjunction with learned indices (Ferragina et al., 2020). Several previous results discuss the trade-off between update time and storage utilization (Brown, 2014; Fagerberg et al., 2019).

# B. Proofs of Treaps

In this section, we formally prove the properties of traditional treaps in Section 2 and the static optimality of our learning-augmented BST in Section 2.2.

**Lemma 2.2** ((Aragon & Seidel, 1989))**.** *Let $U(0,1)$ be the uniform distribution over the real interval $[0,1]$. If priority $\sim$ $U(0,1)^n$, each Treap node $x$ has depth $\Theta(\log_2 n)$ with high probability.*

*Proof.* Notice that $\mathsf{depth}(x)$, the depth of item $x$ in the Treap, is the number of ancestors of $x$ in the Treap. Linearity of expectation yields

$$
\begin{aligned}
\mathbb{E}[\mathsf{depth}(x)] &= \sum_{y\in[n]} \mathbb{E}\left[\mathbf{1}_{y \text{ is an ancestor of } x}\right] \\
&= \sum_{y\in[n]} \Pr\left(y \text{ is an ancestor of } x\right) \\
&= \sum_{y\in[n]} \Pr\left(\mathsf{priority}_y = \max_{z\in[x,y]} \mathsf{priority}_z\right) \\
&= \sum_{y\in[n]} \frac{1}{|x-y+1|} = \Theta(\log_2 n).
\end{aligned}
$$

$\square$

**Theorem 2.8.** *For any item $x \in [n]$, we set its priority as*

$$
\mathsf{priority}_x := -\left\lfloor \log_2 \log_2 \frac{m}{f_x} \right\rfloor + \delta_x, \delta_x \sim U(0,1).
$$

*In the corresponding Treap, each node $x$ has expected depth $O(\log_2(m/f_x))$. Therefore, the total time for processing the access sequence is $O(\sum_x f_x \log_2(m/f_x))$, which matches the performance of the optimal static BSTs up to a constant factor.*

*Proof.* Given item frequencies , we define the following $\boldsymbol{w}$ assignment:

$$
w_x := \frac{f_x}{m}, \ x \in [n]. \tag{4}
$$

One can verify that $\|\boldsymbol{w}\|_1 = O(1)$. By Theorem 2.4, the expected depth of each item $x$ is $O(\log_2(m/f_x))$. $\square$

# C. Learning-Augmented B-Trees

We now extend the ideas above, specifically the composite priority notions, to B-Trees in the *External Memory Model*. We show that the learning-augmented B-Treaps (Appendix C.1) obtain static optimality (Appendix C.2) and is robust to the noisy predicted scores (Appendix C.3). This model is also the basis of our analyses in online settings in Appendix D.

## C.1. Learning-Augmented B-Treaps

We first formalize this extension by incorporating our composite priorities with the B-Treap data structure from (Golovin, 2009) and introducing offsets in priorities.

**Lemma C.1** (B-Treap, (Golovin, 2009))**.** *Given the unique binary Treap $(T, \mathsf{priority}_x)$ over the set of items $[n]$ with their associated priorities, and a target branching factor $B = \Omega(\ln^{1/(1-\alpha)} n)$ for some $\alpha > 0$. Assuming $\mathsf{priority}_x$ are drawn uniformly from $(0,1)$, we can maintain a B-Tree $T^B$, called the B-Treap, uniquely defined by $T$. This allows operations such as Lookup, Insert, and Delete of an item to touch $O(\frac{1}{\alpha}\log_B n)$ nodes in $T^B$ in expectation.*

*In particular, if $B = O(n^{1/2-\delta})$ for some $\delta > 0$, all above performance guarantees hold with high probability.*

The main technical theorem is the following:

**Theorem C.2** (Learning-Augmented B-Treap via Composite Priorities). *Denote $\boldsymbol{w} = (w_1, \cdots, w_n) \in (0,1)^n$ as a score associated with each element of $[n]$ such that $\|\boldsymbol{w}\|_1 = O(1)$ and a branching factor $B = \Omega(\ln^{1/(1-\alpha)} n)$, there is a randomized data structure that maintains a B-Tree $T^B$ over $U$ such that*

1. *Each item $x$ has expected depth $O(\frac{1}{\alpha} \log_B(1/w_x))$.*

2. *Insertion or deletion of item $x$ into/from $T$ touches $O(\frac{1}{\alpha} \log_B(1/w_x))$ nodes in $T^B$ in expectation.*

3. *Updating the weight of item $x$ from $w$ to $w'$ touches $O(\frac{1}{\alpha}|\log_B(w'/w)|)$ nodes in $T^B$ in expectation.*

*We consider the following priority assignment scheme: For any $x$ and its corresponding score $w_x$, we always maintain:*

$$\mathsf{priority}_x := -\lfloor \log_2 \log_B \frac{1}{w_x} \rfloor + \delta, \ \delta \sim U(0,1).$$

*In addition, if $B = O(n^{1/2-\delta})$ for some $\delta > 0$, all above performance guarantees hold with high probability $1 - \delta$.*

The learning-augmented B-Treap is created by applying Lemma C.1 to a partition of the binary Treap $T$. Each item $x$ has a priority in the binary Treap $T$, defined as:

$$\mathsf{priority}_x = -\left\lfloor \log_2 \log_B \frac{1}{w_x} \right\rfloor + \delta_x, \delta_x \sim U(0,1), \ \text{for all } x \in U. \tag{5}$$

We then partition the binary Treap $T$ based on each item's tier. The tier of an item is defined as the absolute value of the integral part of its priority, i.e., $\tau_x \overset{\text{def}}{=} \lfloor \log_2 \log_B(1/w_x) \rfloor$.

*Proof of Theorem C.2.* To formally construct and maintain $T^B$, we follow these steps:

1. Start with a binary Treap $(T, \mathsf{priority}_x)$ with priorities defined using equation (5).

2. Decompose $T$ into sub-trees based on each item's tier, resulting in a set of maximal sub-trees with items sharing the same tier.

3. For each $T_i$, apply Lemma C.1 to maintain a B-Treap $T_i^B$.

4. Combine all the B-Treaps into a single B-Tree, such that the parent of $\mathsf{root}(T_i^B)$ is the B-Tree node containing the parent of $\mathsf{root}(T_i)$.

Now, let's analyze the depth of each item $x$. Keep in mind that any item $y$ in the same B-Tree node shares the same tier. Therefore, we can define the tier of each B-Tree node as the tier of its items.

Suppose $x_1, x_2, \ldots$ are the B-Tree nodes we encounter until we reach $x$. The tiers of these nodes are in non-increasing order, that is, $\tau_{x_i} \geq \tau_{x_{i+1}}$ for any $i$. We'll define $C_t$ as the number of items of tier $t$ for any $t$. As per the definition (refer to equation (5)), we have:

$$C_t = O(B^{2^t}), \ \text{for all } t$$

Using Lemma C.1 and the fact that $B = O(C_t^{1/2}), t \geq 1$, we find that the number of nodes among $x_i$ of tier $t$ is $O(\frac{1}{\alpha} \log_B C_t) = O(2^t/\alpha)$ with high probability. As a result, the number of nodes touched until reaching $x$ is, with high probability:

$$\sum_{t=0}^{\tau_x} O(2^t/\alpha) = O(2^{\tau_x}/\alpha) = O\left(\frac{1}{\alpha} \log_B \frac{1}{w_x}\right)$$

This analysis is also applicable when performing Lookup, Insert, and Delete operations on item $x$.

The number of nodes touched when updating an item's weight can be derived from first deleting and then inserting the item.

$\square$

## C.2. Static Optimality

In this section, we show that with our priority assignment, the learning-augmented B-Treaps are statically optimal. Let $x(1), x(2), \ldots, x(m)$ represent an access sequence of length $m$. We define the relative frequency of each item $x$ as $p_x \stackrel{\text{def}}{=} \frac{|\{i \in [m] \mid x(i) = x\}|}{m}$. A static B-Tree is statically optimal if the depth of item $x$ is:

$$\mathsf{depth}(x) = O\left(\log_B \frac{1}{p_x}\right), \text{ for all } x \in [n]$$

As a corollary of Theorem C.2, we show that if we are given the relative frequency $p_x$, the learning-augmented B-Treaps with our priority assignment achieves *Static Optimality* with weights $w_x \stackrel{\text{def}}{=} p_x$.

**Corollary C.3** (Static Optimality for B-Treaps)**.** *Given the relative frequency $p_x$ of each item $x \in [n]$, and a branching factor $B = \Omega(\ln^{1.1} n)$, there exists a randomized data structure that maintains a B-Tree $T^B$ over $[n]$ such that each item $x$ has an expected depth of $O(\log_B 1/p_x)$. That is, $T^B$ achieves* Static Optimality*, meaning the total number of nodes touched is $O(OPT_B^{static})$ in expectation, where:*

$$OPT_B^{static} \stackrel{\text{def}}{=} m \cdot \sum_{x \in [n]} p_x \log_B \frac{1}{p_x} \tag{6}$$

*Furthermore, if $B = O(n^{1/2-\delta})$ for some $\delta > 0$, all above performance guarantees hold with high probability.*

## C.3. Robustness Guarantees

In practice, we would not have access to the relative frequency $p_x$. Instead, we will have an inaccurate prediction $q_x$. Let $\boldsymbol{p}$ and $\boldsymbol{q}$ be the probability distribution over $[n]$ such that $\boldsymbol{p}(x) = p_x, \boldsymbol{q}(x) = q_x$. In this section, we will show that B-Treap performance is robust to the error. Specifically, we analyze the performance under various notions of error in the prediction. The notions listed here are the ones used for learning discrete distributions (refer to (Canonne, 2020) for a comprehensive discussion).

**Corollary C.4** (Kullback—Leibler (KL) Divergence)**.** *If we are given a density prediction $\boldsymbol{q}$ such that $d_{KL}(\boldsymbol{p}; \boldsymbol{q}) = \sum_x p_x \ln(p_x/q_x) \leq \epsilon$, the total number of touched nodes is*

$$O\left(OPT_B^{static} + \frac{\epsilon m}{\ln B}\right)$$

*Proof.* Given the inaccurate prediction $\boldsymbol{q}$, the total number of touched nodes in $T^B$ is

$$O\left(\sum_x m \cdot p_x \log_B \frac{1}{q_x}\right)$$
$$= O\left(\sum_x m \cdot p_x \log_B \frac{1}{p_x} + m \cdot \sum_x p_x \log_B \frac{p_x}{q_x}\right)$$
$$= O(OPT_B^{static} + m \frac{d_{KL}(\boldsymbol{p}; \boldsymbol{q})}{\ln B})$$

$\square$

**Corollary C.5** ($\chi^2$)**.** *If we are given a density prediction $\boldsymbol{q}$ such that $\chi^2(\boldsymbol{p}; \boldsymbol{q}) = \sum_x (p_x - q_x)^2/q_x \leq \epsilon$, the total number of touched nodes is*

$$O\left(OPT_B^{static} + \frac{\epsilon m}{\ln B}\right)$$

*Proof.* The corollary follows from Corollary C.5 and the fact $d_{KL}(\boldsymbol{p}; \boldsymbol{q}) \leq \chi^2(\boldsymbol{p}; \boldsymbol{q}) \leq \epsilon$. $\square$

**Corollary C.6** ($L_\infty$ Distance)**.** *If we are given a density prediction $\boldsymbol{q}$ such that $\|\boldsymbol{p} - \boldsymbol{q}\|_\infty \leq \epsilon$, the total number of touched nodes is*

$$O\left(OPT_B^{static} + m \log_B(1 + \epsilon n)\right)$$

*Proof.* For item $x$ with its marginal probability smaller than $1/1000n$, its expected depth in the B-Treap is $O(\log_B n)$ using either $p_x$ or $q_x$ as its score. If item $x$'s marginal probability is at least $1/1000n$, the $L_\infty$ distance implies that

$$\frac{p_x}{\widetilde{p}_x} = 1 + \frac{p_x - \widetilde{p}_x}{\widetilde{p}_x} \leq 1 + \frac{\epsilon}{1/(1000n)} = 1 + 1000(1 + \epsilon n)$$

Therefore, item $x$'s expected depth in the B-Treap with score $\boldsymbol{q}$ is roughly

$$O\left(\log_B \frac{1}{p_x} + \log_B \frac{p_x}{q_x}\right) \leq O\left(\log_B \frac{1}{p_x} + \log_B(1 + \epsilon n)\right)$$

The corollary follows. □

**Corollary C.7** ($L_2$ Distance). *If we are given a density prediction $\boldsymbol{q}$ such that $\|\boldsymbol{p} - \boldsymbol{q}\| \leq \epsilon$, the total number of touched nodes is*

$$O\left(OPT_B^{static} + m \log_B(1 + \epsilon n)\right)$$

*Proof.* This claim follows from Corollary C.6 and the fact $\|\boldsymbol{p} - \boldsymbol{q}\|_\infty \leq \|\boldsymbol{p} - \boldsymbol{q}\|_2 \leq \epsilon$. □

**Corollary C.8** (Total Variation). *If we are given a density prediction $\boldsymbol{q}$ such that $d_{TV}(\boldsymbol{p}, \boldsymbol{q}) = 0.5\|\boldsymbol{p} - \boldsymbol{q}\|_1 \leq \epsilon$, the total number of touched nodes is*

$$O\left(OPT_B^{static} + m \log_B(1 + \epsilon n)\right)$$

*Proof.* This claim follows from Corollary C.6 and the fact $\|\boldsymbol{p} - \boldsymbol{q}\|_\infty \leq \|\boldsymbol{p} - \boldsymbol{q}\|_1 \leq 2\epsilon$. □

**Corollary C.9** (Hellinger Distance). *If we are given a density prediction $\boldsymbol{q}$ such that $d_H(\boldsymbol{p}, \boldsymbol{q}) = 0.5\|\sqrt{\boldsymbol{p}} - \sqrt{\boldsymbol{q}}\|_2 \leq \epsilon$, the total number of touched nodes is*

$$O\left(OPT_B^{static} + m \log_B(1 + \epsilon n)\right)$$

*Proof.* This claim follows from Corollary C.6 and the fact $\|\boldsymbol{p} - \boldsymbol{q}\|_\infty \leq 2\sqrt{2}d_H(\boldsymbol{p}, \boldsymbol{q}) \leq 2\sqrt{2}\epsilon$. □

## D. Dynamic Learning-Augmented Search Trees

In this section, we investigate the properties of dynamic B-trees that permit modifications concurrent with sequence access. Prioritizing items that are anticipated to be accessed in the near future to reside at lower depths within the tree can significantly reduce access times. Nonetheless, updating the B-trees introduces additional costs. The overarching goal is to minimize the composite cost, which includes both the access operations across the entire sequence and the modifications to the B-trees. We specifically concentrate on the study of locally dynamic B-trees, which are characterized by the restriction that tree modifications are limited solely to the adjustment of priorities for the items being accessed.

In Appendix D.1, we give the total cost guarantees for any locally dynamic B-trees. In Appendix D.2, we establish the robustness guarantees in the context of imprecise priority scores, which may be given from a learning oracle. In Appendix D.3, we demonstrate that the dynamic learning-augmented B-trees with a specific priority based on the working set size — the number of distinct items requested between two consecutive accesses — achieves the working set property. Full details are included in the appendix. Finally, in Appendix D.4, we analyze the general dynamic B-trees with general time-varying priorities.

### D.1. Locally Dynamic B-trees

Our objective is to maintain a data structure that minimizes the total cost of accessing the sequence $S$ given the time-varying score $\boldsymbol{w}(i) \in (0, 1)^n, i \in [m]$ associated to each item. Here, we focus on the dynamic B-trees that update the priorities of only the items being accessed at any given moment, leaving the priorities of all other items unchanged. We refer to these as *locally dynamic B-trees.*

Given $n$ items, denoted as $[n] = \{1, \cdots, n\}$, and a sequence of access sequence $\boldsymbol{X} = (x(1), \ldots, x(m))$, where $x(i) \in [n]$. At time $i \in [m]$, there exists some time-dependent score $w_{ij}$ associated with each item $j \in [n]$. Let $\boldsymbol{w}(i) = (w_{i,1}, \cdots, w_{i,n}) \in (0,1]^n$ be the time-varying score vector. The score $\boldsymbol{w}(i)$ is defined to be *locally changed* if it only differs from the previous score vector at the index of the item being accessed. In other words, at each time $i \in \{1, 2, \cdots, n\}$, we have $w_{i,j} = w_{i-1,j}$ for any $j \neq x(i)$. The *locally dynamic B-Treap* is then defined as a B-Treap whose priorities are updated according to the locally changed score. For any vector $\boldsymbol{w}$, we write $\log \boldsymbol{w}$ as the vector taking the element-wise $\log$ on $\boldsymbol{w}$. We give the guarantees in Theorem D.1.

**Theorem D.1** (Locally-Dynamic B-Treap with Given Priorities). *Given the locally changed scores $\boldsymbol{w}(i) \in (0,1)^n, i \in [m]$ satisfying $\|\boldsymbol{w}(i)\|_1 = O(1)$ and a branching factor $B = \Omega(\ln^{1.1} n)$, there is a randomized data structure that maintains a B-Tree $T^B$ over $[n]$ such that when accessing the item $x(i)$ at time $i$, the expected depth of item $x(i)$ is $O(\log_B \frac{1}{w_{i,x(i)}})$. The expected total cost for processing the whole access sequence $\boldsymbol{X}$ is*

$$\text{cost}(\boldsymbol{X}, \boldsymbol{w}) = O\left(n \log_B n + \sum_{i=1}^{m} \log_B \frac{1}{w_{i,x(i)}}\right).$$

*Moreover, if $B = O(n^{1/2-\delta})$ for some $\delta > 0$, the guarantees hold with probability $1 - \delta$.*

The proof is an application of Theorem C.2, where the priority function dynamically changes as time goes on, rather than the *Static Optimality* case where the priority is fixed beforehand.

*Proof of Theorem D.1.* Initially, we set the priority for all items to be 1, and insert all items into the Treap. For any time $i \in [n]$, for $j \in [n]$ such that $w_{i-1,j} \neq w_{i,j}$, we set

$$\text{priority}_j^{(i)} := -\lfloor \log_4 \log_B \frac{1}{w_{i,j}} \rfloor + \delta_{ij}, \delta_{ij} \sim U(0,1).$$

Since $\|w(i)\|_1 = O(1), i \in [m]$, by Theorem C.2, the expected depth of item $s(i)$ is $O(\log_B \frac{1}{w_{i,x(i)}})$. The total cost for processing the sequence consists of both accessing $x(i)$ and updating the priorities. The expected total cost for all the accesses is

$$O\left(\sum_{i=1}^{m} \log_B \frac{1}{w_{i,x(i)}}\right).$$

Then we will calculate the cost to update the Treap. Since the priority of an item only changes when it is accessed. Updating the priority of $x(i)$ from $w_{i-1,x(i)}$ to $w_{i,x(i)}$ has cost

$$O\left(\left|\log_B \frac{w_{i-1,x(i)}}{w_{i,x(i)}}\right|\right).$$

Hence we can bound the expected total cost for maintaining the Treap by

$$O\left(n \log_B n + \sum_{i=2}^{m} \left|\log_B \frac{w_{i-1,x(i)}}{w_{i,x(i)}}\right|\right)$$

$$= O\left(n \log_B n + 2\sum_{i=1}^{m} \log_B \frac{1}{w_{i,x(i)}}\right).$$

Together the expected total cost is

$$O\left(n \log_B n + \sum_{i=1}^{m} \log_B \frac{1}{w_{i,x(i)}}\right).$$

The high probability bound follows similarly as Theorem C.2. $\qquad \square$

## D.2. Robustness Guarantees

We have shown that given time-varying scores associated with each item $\boldsymbol{w}(i)$, there exists a B-Tree that gives us the total cost in terms of the scores. In this section, we address scenarios in which precise score $\boldsymbol{w}(i)$ are not accessible. Utilizing a learning oracle that predicts the logarithm of the score, we demonstrate that the total cost incorporates an additive term corresponding to the mean absolute error (MAE) of the logarithm of the scores $\sum_{i=1}^{m} |\log_B w_{i,x(i)} - \log_B \widetilde{w}_{i,x(i)}|$. We predict the logarithm of the score instead of itself to better capture the scale of it.

**Theorem D.2** (Locally Dynamic B-Treap with Predicted Scores). *Given the predicted locally changed scores $\widetilde{\boldsymbol{w}}(i) \in (0,1)^n$ satisfying $\|\widetilde{\boldsymbol{w}}(i)\|_1 = O(1)$, $\widetilde{w}_{i,j} \geq 1/\text{poly}(n)$ and a branching factor $B = \Omega(\ln^{1.1} n)$, there is a randomized data structure that maintains a B-Tree over the $n$ keys such that the expected total cost for processing the whole access sequence $\boldsymbol{X}$ is*

$$\text{cost}(\boldsymbol{X}, \widetilde{\boldsymbol{w}}) = \text{cost}(\boldsymbol{X}, \boldsymbol{w}) + O\left(\sum_{i=1}^{m} \left|\log_B w_{i,x(i)} - \log_B \widetilde{w}_{i,x(i)}\right|\right).$$

*Moreover, if $B = O(n^{1/2-\delta})$ for some $\delta > 0$, the guarantees hold with probability $1 - \delta$.*

*Proof.* We apply Theorem D.1 with the predicted score $\widetilde{\boldsymbol{w}}(i)$, and get the expected total loss is

$$\text{cost}(\boldsymbol{X}, \widetilde{\boldsymbol{w}}) = O\left(n \log_B n + \sum_{i=1}^{m} \log_B \frac{1}{\widetilde{w}_{i,x(i)}}\right)$$

$$\leq \text{cost}(\boldsymbol{X}, \boldsymbol{w}) + O\left(\sum_{i=1}^{m} \left|\log_B \frac{1}{\widetilde{w}_{i,x(i)}} - \log_B \frac{1}{w_{i,x(i)}}\right|\right)$$

$$= \text{cost}(\boldsymbol{X}, \boldsymbol{w}) + O\left(\sum_{i=1}^{m} \left|\log_B w_{i,x(i)} - \log_B \widetilde{w}_{i,x(i)}\right|\right).$$

$\square$

## D.3. Working Set Property

In data structures, the working set is the collection of data that a program uses frequently over a given period. This concept is important because it helps us understand how a program interacts with memory and thus enables us to design more efficient data structures and algorithms. For example, if a program is sorting a list, the working set might be the elements of the list it is comparing and swapping right now. The size of the working set can affect how fast the program runs. A smaller working set can make the program run faster because it means the program doesn't need to reach out to slower parts of memory as often. In other word, if we know which parts of a data structure are used most, we can organize the data or even the memory in a way that makes accessing these parts faster, which can speed up the entire program.

In this section, we construct dynamic learning-augmented B-treaps that achieve the working set property. We define the *working-set size* as the number of distinct items accessed between two consecutive accesses. Correspondingly, we design a time-varying score, *working-set score*, as the reciprocal of the square of one plus working-set size. We will show that the working-set score is locally changed and there exists a data structure that achieves the *working-set property*, which states that the time to access an element is a logarithm of its working-set size. The formal definition of the working-set size and the main theorems in this section are presented as follows.

**Definition D.3** (Previous and Next Access $\text{prev}(i,x)$ and $\text{next}(i,x)$). Let $\text{prev}(i,x)$ be the previous access of item $x$ at or before time $i$, i.e, $\text{prev}(i,x) := \max\{i' \leq i \mid x(i') = x\}$. Let $\text{next}(i,x)$ to be the next access of item $x$ after time $i$, i.e, $\text{next}(i,x) := \min\{i' > i \mid x(i') = x\}$.

**Definition D.4** (Working-set Size $\text{work}(i,x)$). Define the *working-set Size* $\text{work}(i,x)$ to be the number of distinct items accessed between the previous access of item $x$ at or before time $i$ and the next access of item $x$ after time $i$. That is,

$$\text{work}(i,x) \overset{\text{def}}{=} |\{x(\text{prev}(i,x) + 1), \cdots, x(\text{next}(i,x))\}|.$$

If $x$ does not appear after time $i$, we define $\text{work}(i,x) := n$.

**Theorem D.5** (Dynamic B-Treaps with Working-set Priority)**.** *With the working-set size* $\mathsf{work}(i,x)$ *known and the branching factor* $B = \Omega(\ln^{1.1} n)$*, there is a randomized data structure that maintains a B-Tree* $T^B$ *over* $[n]$ *with the priorities assigned as*

$$\mathsf{priority}(i,x) = -\lfloor \log_2 \log_B (1 + \mathsf{work}(i,x))^2 \rfloor + U(0,1).$$

*Upon accessing the item* $x$ *at time* $i$*, the expected depth of item* $x$ *is* $O(\log_B(1 + \mathsf{work}(i,x))$*. The expected total cost for processing the whole access sequence* $\boldsymbol{X}$ *is*

$$\mathsf{cost}(\boldsymbol{X}, \mathsf{priority}) = O\left( n \log_B n + \sum_{i=1}^{m} \log_B(1 + \mathsf{work}(i,x)) \right)$$

*In particular, if* $B = O(n^{1/2-\delta})$ *for some* $\delta > 0$*, the guarantees hold with probability* $1 - \delta$*.*

**Remark.** Consider two sequences with length $m$, $\boldsymbol{X}_1 = (1, 2, \cdots, n, 1, 2, \cdots, n, \cdots, 1, 2, \cdots, n)$, $\boldsymbol{X}_2 = (1, 1, \cdots, 1, 2, 2, \cdots, 2, \cdots, n, n, \cdots, n)$. Two sequences have the same total cost if we have a fixed score. However, $X_2$ should have less cost because of its repeated pattern. Given the frequency freq as a time-invariant priority, by Corollary C.3, the optimal static costs are

$$\mathsf{cost}(\boldsymbol{X}_1, \mathsf{freq}) = \mathsf{cost}(\boldsymbol{X}_2, \mathsf{freq}) = O(m \log_B n).$$

But for the dynamic B-Trees, with the working-set score, we calculate both costs from Theorem D.5 as

$$\mathsf{cost}(\boldsymbol{X}_1, \omega) = O(m \log_B(n+1)),$$
$$\mathsf{cost}(\boldsymbol{X}_2, \omega) = O(n \log_B n + m \log_B 3).$$

This means that our proposed priority can better capture the timing pattern of the sequence and thus can even do better than the optimal static setting.

The main idea to prove Theorem D.5 is to show that (1) the working-set size is locally changed and (2) the corresponding priority satisfies the regularity conditions in Theorem D.1. To complete the proof, we introduce the interval-set size $\mathsf{interval}(i,x)$. See Figure 7 as an illustration.

**Definition D.6** (Interval-set Size $\mathsf{interval}(i,x)$)**.** Define the *Interval-set Size* $\mathsf{interval}(i,x)$ to be the number of distinct items accessed between time $i$ and the *next* access of item $x$ after time $i$. That is,

$$\mathsf{interval}(i,x) := |\{x(i+1), \cdots, x(\mathsf{next}(i,x))\}|.$$

If $x$ does not appear after time $i$, we define $\mathsf{interval}(i,x) := n$.

Furthermore, we define the working-set score as follows.

**Definition D.7** (Working-set Score $\omega(i,x)$)**.** Define the time-varying priority as the reciprocal of the square of one plus working-set size. That is,

$$\omega(i,x) = \frac{1}{(1 + \mathsf{work}(i,x))^2}$$

Next, we will show that the interval set priority is $O(1)$ for any time $i \in [m]$ in Lemma D.8. The proof has three steps. Firstly, the interval-set size at time $i$ is always a permutation of $[n]$. Secondly, for any $i \in [m], x \in [n]$, the working-set size is always no less than the interval-set size. Therefore, for any $i \in [m]$, the $l_1$ norm of working-set score vector $\omega(i) \stackrel{\text{def}}{=} (\omega(i,1), \cdots, \omega(i,n))$ can be upper bounded by $\sum_{j=1}^{n} 1/(1+j)^2 = O(1)$.

**Lemma D.8** (Norm Bound for Working-set Score)**.** *Fix any timestamp* $i \in [m]$*,*

$$\sum_{j=1}^{n} \omega(i,j) = O(1).$$

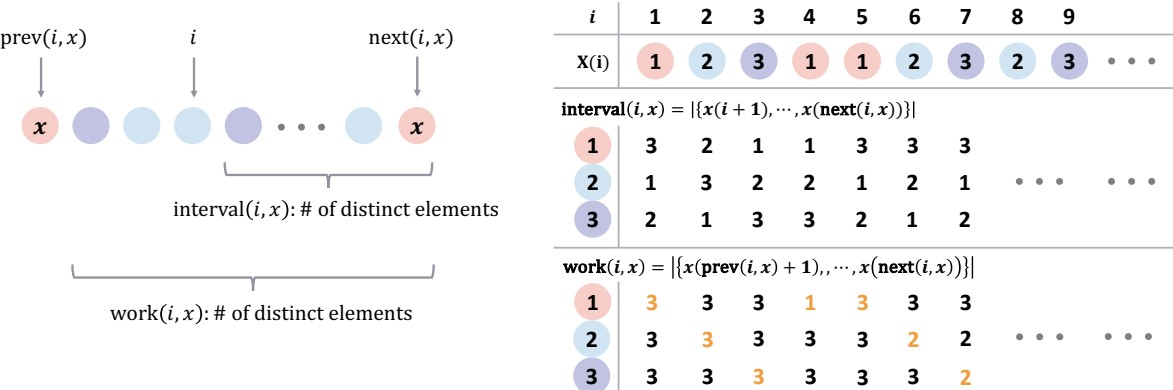

*Figure 7.* An example of $n = 3$, $\boldsymbol{X} = (1, 2, 3, 1, 1, 2, 3, \cdots)$. For any $i, x$, work($i$) is a permutation of $[n]$; interval($i, x$) $\geq$ work($i, x$); interval($i, x$) changes only when $x(i) = x$ (highlighted in orange).

*Proof.* We first show that at any time $i \in [m]$, the interval-set size is a permutation of $[n]$. By definition, interval($i, x$) is the number of items $y$ such that next($i, y$) $\leq$ next($i, x$). Let $\pi_1, \cdots, \pi_n$ be a permutation of all items $[n]$ in the order of increasing next($i, x$). Then for any item $x$, interval($i, x$) is the index of $x$ in $\pi$. So the sum of the reciprocal of squared interval($i, x$) is upper bounded by

$$\sum_{j=1}^{n} \frac{1}{(1 + \mathsf{interval}(i, j))^2} = \sum_{j=1}^{n} \frac{1}{(1 + j)^2} = \Theta(1).$$

Secondly, recall that prev($i, x$) $\leq i$, and hence for any $i \in [m]$, $x \in [n]$, work($i, x$) $\geq$ interval($i, x$). So we have the upper bound for working-set score as follows.

$$\sum_{j=1}^{n} \omega(i, j) = \sum_{j=1}^{n} \frac{1}{(1 + \mathsf{work}(i, x))^2}$$
$$\leq \sum_{j=1}^{n} \frac{1}{(1 + \mathsf{interval}(i, j))^2}$$
$$= O(1).$$

$\square$

Since we have shown the working-set score has constant $l_1$ norm and in each timestamp, we only update one item's priority. We are ready to prove and show the efficiency of the corresponding B-Treap.

*Proof of Theorem D.5.* By Lemma D.8, we know $\|\omega(i)\| = O(1)$, for all $i \in [m]$. Also by definition of work($i, x$), for any $x \in [n]$, work($i - 1, x$) $\neq$ work($i, x$) only when $x(i) = x$. So for each time $i$, at most one item (i.e., $x(i)$) changes its priority. So we apply Theorem D.1 with $w_{i,j} = \omega(i, x), i \in [m], x \in [n]$, and get the total cost

$$\mathsf{cost}(\boldsymbol{X}, \omega) = O\left( n \log_B n + \sum_{i=1}^{m} \log_B(1 + \mathsf{work}(i, x(i))) \right)$$

$\square$

Furthermore, we use the following theorem to show the robustness of the results when the scores are inaccurate. This is a direct corollary of Theorem D.2.

**Theorem D.9** (Locally-Dynamic B-Treaps with Predictions)**.** *Given the predicted locally changed working-set score* $\widetilde{\omega}(i) \in (0, 1)^n$ *satisfying* $\|\widetilde{\omega}(i)\|_1 = O(1)$, $\widetilde{\omega}_{i,j} \geq 1/\mathrm{poly}(n)$ *and the branching factor* $B = \Omega(\ln^{1.1} n)$, *there is a*

*randomized data structure that maintains a B-Tree over the $n$ keys such that the expected total cost for processing the whole access sequence $\boldsymbol{X}$ is*

$$\mathrm{cost}(\boldsymbol{X}, \widetilde{\omega}) = \mathrm{cost}(\boldsymbol{X}, \omega) + O\left(\sum_{i=1}^{m} \left|\log_B \omega_{i,x(i)} - \log_B \widetilde{\omega}_{i,x(i)}\right|\right).$$

*In particular, if $B = O(n^{1/2-\delta})$ for some $\delta > 0$, the guarantees hold with probability $1 - \delta$.*

### D.4. General Results for Dynamic B-Trees

In this section, we give the results for general dynamic B-trees. We first construct the dynamic B-Treaps and give the guarantees when we have access to the real-time priorities for each item in Appendix D.4.1. Then we analyze the dynamic B-trees given the estimation the time-varying priorities in Appendix D.4.2. We use the same notation in Appendix D.

D.4.1. DYNAMIC B-TREAP WITH GIVEN PRIORITIES

**Theorem D.10** (Dynamic B-Treap with Given Priorities). *Given the time-varying scores $\boldsymbol{w}(i) \in (0, 1)^n, i \in [m]$ satisfying $\|\boldsymbol{w}(i)\|_1 = O(1)$ and a branching factor $B = \Omega(\ln^{1.1} n)$, there is a randomized data structure that maintains a B-Tree $T^B$ over $[n]$ such that when accessing the item $x(i)$ at time $i$, the expected depth of item $x(i)$ is $O(\log_B \frac{1}{w_{i,x(i)}})$. The expected total cost for processing the whole access sequence $\boldsymbol{X}$ is*

$$\mathrm{cost}(\boldsymbol{X}, \boldsymbol{w}) = O\left(n \log_B n + \sum_{i=1}^{m} \log_B \frac{1}{w_{i,x(i)}} + \sum_{i=2}^{m} \sum_{j=1}^{n} \left|\log_B \frac{1}{w_{i,j}} - \log_B \frac{1}{w_{i-1,j}}\right|\right).$$

*In particular, if $B = O(n^{1/2-\delta})$ for some $\delta > 0$, the guarantees hold with probability $1 - \delta$.*

*Proof.* Initially, we set the priority for all items to be 1, and insert all items into the Treap. For any time $i \in [n]$, for $j \in [n]$ such that $w_{i-1,j} \neq w_{i,j}$, we set

$$\mathrm{priority}_j^{(i)} := -\lfloor \log_4 \log_B \frac{1}{w_{i,j}} \rfloor + \delta_{ij}, \delta_{ij} \sim U(0, 1).$$

Since $\|w(i)\|_1 = O(1), i \in [m]$, by Theorem C.2, the expected depth of item $s(i)$ is $O(\log_B \frac{1}{w_{i,x(i)}})$. The total cost for processing the sequence consists of both accessing $x(i)$ and updating the priorities. The expected total cost for all the accesses is

$$O\left(\sum_{i=1}^{m} \log_B \frac{1}{w_{i,x(i)}}\right).$$

Then we will calculate the cost to update the Treap. Updating the priority of $j$ from $w_{i-1,j}$ to $w_{i,j}$ has cost $O(|\log_B(w_{i-1,j}/w_{i,j})|)$. Hence we can bound the expected total cost for maintaining the Treap by

$$O\left(n \log_B n + \sum_{i=2}^{m} \sum_{j=1}^{n} \left|\log_B \frac{w_{i-1,j}}{w_{i,j}}\right|\right).$$

Together the expected total cost is

$$O\left(n \log_B n + \sum_{i=1}^{m} \log_B \frac{1}{w_{i,x(i)}} + \sum_{i=2}^{m} \sum_{j=1}^{n} \left|\log_B \frac{1}{w_{i,j}} - \log_B \frac{1}{w_{i-1,j}}\right|\right).$$

The high probability bound follows similarly as Theorem C.2. $\qquad \square$

**Remark.** The total cost for processing the access sequence has three terms. The first two terms are the same as in the static optimality bound, while the third term is incurred from updating the scores. Hence, here is a trade-off between the costs of updating items and the benefits from the time-varying scores. Moreover, the locally-dynamic B-trees can avoid the high cost of keeping updating the scores because only one score is changed per time.

### D.4.2. DYNAMIC B-TREAP WITH PREDICTED PRIORITIES

In this section, we give the guarantees for the dynamic B-Treaps with predicted priorities learned by a machine learning oracle. Similar as in Appendix D.4.2, we here predict $\log_B \frac{1}{w_{i,j}}$ to better capture the scale of the scores. And we will find that the total cost using the B-Trees using the predicted scores is equal to the cost using the accurate priorities plus an additive error that is linear in the mean absolute error of our prediction scores:

$$\sum_{i=1}^{m} \sum_{j=1}^{n} \left| \log_B \frac{1}{w_{i,j}} - \log_B \frac{1}{\widetilde{w}_{i,j}} \right|.$$

**Theorem D.11** (Dynamic B-Treap with Predicted Scores). *Given the predicted time-varying scores $\widetilde{\boldsymbol{w}}(i) \in (0,1)^n$ satisfying $\|\widetilde{\boldsymbol{w}}(i)\|_1 = O(1)$, $\widetilde{w}_{i,j} \geq 1/\mathrm{poly}(n)$ and a branching factor $B = \Omega(\ln^{1.1} n)$, there is a randomized data structure that maintains a B-Tree over the $n$ keys such that the expected total cost for processing the whole access sequence $\boldsymbol{X}$ is*

$$\mathrm{cost}(\boldsymbol{X}, \widetilde{\boldsymbol{w}}) = \mathrm{cost}(\boldsymbol{X}, \boldsymbol{w}) + O\left( \sum_{i=1}^{m} \sum_{j=1}^{n} \left| \log_B \frac{1}{w_{i,j}} - \log_B \frac{1}{\widetilde{w}_{i,j}} \right| \right)$$

*In particular, if $B = O(n^{1/2-\delta})$ for some $\delta > 0$, the guarantees hold with probability $1 - \delta$.*

*Proof.* We apply Theorem D.10 with score $\widetilde{w}$, and get the expected depth of $x(i)$ is

$$O\left( \log_B \frac{1}{\widetilde{w}_{i,j}} \right).$$

The expected total cost is

$$
\begin{aligned}
\mathrm{cost}(\boldsymbol{X}, \widetilde{\boldsymbol{w}}) =& O\left( n \log_B n + \sum_{i=1}^{m} \log_B \frac{1}{\widetilde{w}_{i,x(i)}} + \sum_{i=2}^{m} \sum_{j=1}^{n} \left| \log_B \frac{1}{w_{i,j}} - \log_B \frac{1}{\widetilde{w}_{i-1,j)}} \right| \right) \\
=& \mathrm{cost}(\boldsymbol{X}, \boldsymbol{w}) + O\left( \sum_{i=1}^{m} \left| \log_B \frac{1}{w_{i,x(i)}} - \log_B \frac{1}{\widetilde{w}_{i,x(i)}} \right| \right) \\
& + O\left( \sum_{i=2}^{m} \sum_{j=1}^{n} \left| \log_B \frac{1}{w_{i,j}} - \log_B \frac{1}{\widetilde{w}_{i,j}} \right| + \sum_{i=1}^{m-1} \sum_{j=1}^{n} \left| \log_B \frac{1}{w_{i,j}} - \log_B \frac{1}{\widetilde{w}_{i,j}} \right| \right) \\
=& \mathrm{cost}(\boldsymbol{X}, \boldsymbol{w}) + O\left( \sum_{i=1}^{m} \sum_{j=1}^{n} \left| \log_B \frac{1}{w_{i,j}} - \log_B \frac{1}{\widetilde{w}_{i,j}} \right| \right)
\end{aligned}
$$

$\square$

## E. Additional Experiments on Inaccurate Prediction Oracle

### E.1. Mixture of Distributions

We consider a setting where the actual data follows a mixture of two distributions while the frequency predictor provides only a single distribution. Specifically, we assume that the predicted frequency follows the adversarial distribution, whereas the actual access sequence is generated by a mixture of two distributions: with probability $w$, the item follows the adversarial distribution and with probability $1 - w$, it follows the Zipfian distribution (Figure 8a, Figure 8c, Figure 8e) or uniform distribution (Figure 8b, Figure 8d, Figure 8f). We set $n = 1000$ and vary $m$ over $[2000, 6000, 10000, 16000, 20000]$. The $x$-axis represents the number of unique items, and the $y$-axis denotes the number of comparisons made, which measures access cost. We compare our Treaps against Splay trees and randomized Treaps. Our experiments show that our Treaps outperform both alternatives, even when $75\%$ of the data comes from an unknown distribution (either Zipfian or uniform). Furthermore, as the prediction quality decreases, the performance of our Treaps remains stable, demonstrating their robustness.

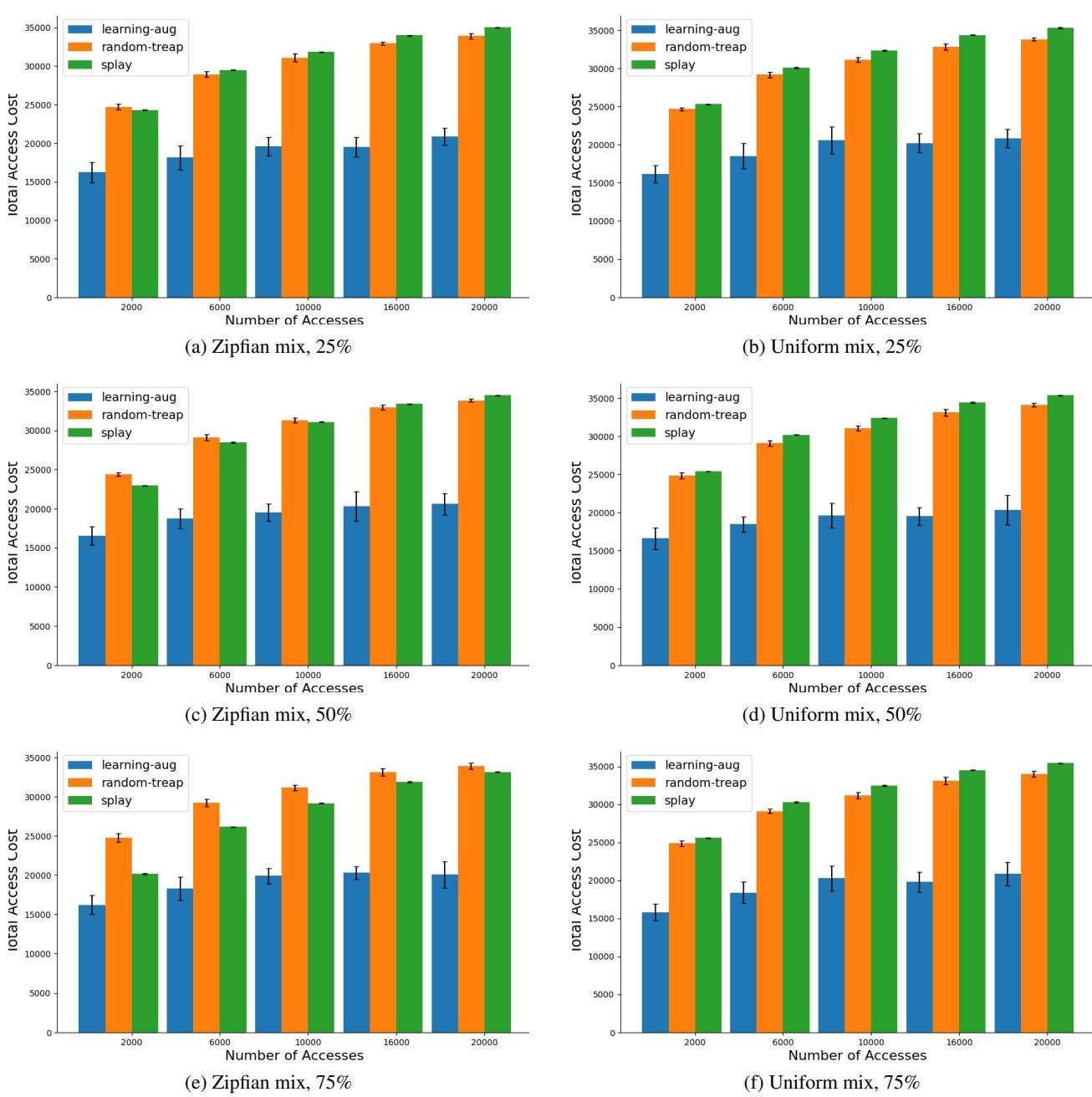

Figure 8. Error rates under mixture distributions. Left column: Zipfian mixing; Right column: uniform mixing.

