# OpenReview forum: "On the Power of Learning-Augmented Search Trees"
_ICML.cc/2025/Conference — ICML 2025 poster_

### Official Review · Reviewer_R8jP · 2025-03-09

**Overall Recommendation:** 4

**Summary:**

This paper examines the integration of external predictions into classic binary search trees (BSTs) and B-Trees via Treaps. Prior attempts to build such a data structure only had guarantees for Zipfian distributions and were shown to be provably bad on certain sequences of inputs. In this work, the authors propose a composite priority functions to balance nodes according to predicted access frequencies and proved optimality in the static setting and guarantees with respect to working sets in the dynamic setting when the tree is allowed to adapt to the input sequence. Experiments were also done to empirically evaluate the newly proposed method, showing improving over existing solutions.

**Claims And Evidence:**

Yes

**Essential References Not Discussed:**

Not that I am aware of

**Ethical Review Concerns:**

NIL

**Experimental Designs Or Analyses:**

I checked the experimental section in the main paper

**Methods And Evaluation Criteria:**

Yes

**Other Comments Or Suggestions:**

- Typo: On Page 2, paragraph on Static Optimality of Learning-Augmented Search Trees, "... Equation Equation (2)..." should be "... Equation (2)..."

**Other Strengths And Weaknesses:**

# Strengths

- The proposed composite priority scheme enabled the authors to obtain guarantees for arbitrary input distributions, beyond just Zipfian.
- The proposed method has nice provable guarantees such as static optimality, dynamic adaptability, and error robustness.
- Experiments were also done to empirically evaluate the newly proposed method, showing improving over existing solutions.
# Weaknesses

- Typically, in the learning-augmented literature, the "robustness" guarantee is ideally asymptotically similar to the setting without predictions. The authors gave a robustness bound in terms of KL divergence between the true frequencies and predicted frequencies. I was hoping the authors can discuss and relate their bound with the setting without predictions.
- While I can imagine how to get frequency predictions for the static setting, it is unclear how one can obtain the predictions needed for the dynamic setting to work well. I hope that the authors can point me to a discussion that I might have overlooked or mention what explanation they will add in their revision.

**Questions For Authors:**

- Clarification: In the proof plan on Page 4, the expected depth of x refers to the number of its ancestors, not just in S_t, right? If so, in the proof of Theorem 2.4 on Page 5, what about the ancestors of x outside of S_t? What am I misunderstanding?
- I might have missed it but could you describe how the B-tree is being updated in the dynamic setting?

**Relation To Broader Scientific Literature:**

This work contributes to the field of learning-augmented algorithms in particular the development of data structures with predictions.

**Theoretical Claims:**

I checked the proofs in the main paper

---

> ### Author Rebuttal · Authors · 2025-03-30
>
> Thank you for your thoughtful comments! We address the questions and concerns as follows.
>
> **Robustness**
> If no prediction is available, we can naturally set the "predicted frequency" to be $1/n$. In this case, the access cost becomes $O(\log n)$.
>
>
> **Predictions for Dynamic Settings**
> Thank you for highlighting this point! We will include a paragraph in the revision regarding the prediction of working-set scores.
>
> > The working-set size (Definition 4.2) captures the temporal locality and diversity of user behavior around item $x$. Although it requires knowledge of future access items, it aligns conceptually with practices in recommendation systems, where temporal context is used to model user intent and predict relevance. For example, session-based recommendation often considers the diversity of items within a user's recent session [1]. In practice, we can approximate this score using causal proxies (e.g., past-only working-set size) or apply machine learning models to predict it based on observable access patterns.
>
>
> **Clarification on Proof Plan**
> The priority score of $x$ consists of the tier $\tau_x$, which is an integer, and $\delta_x \in (0,1)$. Consequently, all ancestors of the node $x$ must have tier less than or equal to $\tau_x$, and thus are part of the set $S_t$ for $t\leq \tau_x$. In other words, there are no ancestors of $x$ outside the sets $S_t$.
>
> **B-tree Update**
> At each time step, we first compute the working-set score of the current item, then remove the item from the B-tree, and finally re-insert the node into the B-tree with the updated score accordingly. Note that B-Treaps support insertions and deletions efficiently [Lemma C.1].
>
> [1] Wang, Shoujin, et al. "A survey on session-based recommender systems." ACM Computing Surveys (CSUR) 54.7 (2021): 1-38.

---

> > ### Comment · Reviewer_R8jP · 2025-04-01
> >
> > Thank you for the clarifications. I maintain my score as it is.

---

### Official Review · Reviewer_tuW9 · 2025-03-10

**Overall Recommendation:** 4

**Summary:**

This paper is concerned with learning-augmented search trees. Given a dataset of a sequence of requested keys, the recent work by Lin et al. 2022 constructs a learning-augmented search tree as a treap where the "priority" $p(x)$ of each key $x$ is set to be equal to the frequency $f_x$ of the key in the sequence. For simplicity, we will assume that the keys are being drawn from a distribution $p$, and that the empirical frequency in the sequence is exactly equal to the probability of each key. A classical lower bound result due to Mehlhorn 1975 shows that for any tree, the expected cost to access a key drawn from the distribution is at least $\Omega(H(p))$, where $H(p)$ is the Shannon entropy of $p$. If $p$ is the Zipfian distribution, then the treap constructed by Lin et al. 2022 attains this optimal cost. However, their construction was not known to be competitive against the static optimal tree for general distributions. In fact, one of the results of the authors (Theorem 2.14) constructs a distribution for which the cost is indeed suboptimal.

The first main result in the paper is a construction of a treap corresponding to a different priority assignment to the keys. The authors set $p(x) = \log\log p_x + \delta_x$, where $\delta_x$ is uniformly random in $(0,1)$. The authors show (Theorem 2.8) that this priority assignment results in a treap, for which the expected cost of accessing a new key meets the lower bound above for every distribution. Furthermore, even if we only have access to estimates $q_x$ of $p_x$ that are noisy, and build a treap according to the priority function using these noisy estimates, the authors show that the expected cost of accessing a key only increases by the KL divergence between $p$ and $q$. This is a natural characterization of the robustness of their construction.

The next results in the paper are an extension of the above results from binary trees to arbitrary B-trees that have an arity of $B>2$. The authors establish similar results (Theorem 3.1) as the binary case for B-trees as well.

The authors also consider a dynamic setting where the priority function (and hence the maintained treap) is allowed to be changed as we see new keys. In this setting, the authors show a way of constructing dynamic priority functions, such that the resulting dynamic treap attains a "working-set bound", which is a natural benchmark for dynamic data structures.

Finally, the authors perform experiments on synthetic data, comparing their treap data structure to other learning augmented search trees, as well as classical data structures, and show that their data structure compares much better to these other methods.

## update after rebuttal
I thank the authors for their response. The clarifications are useful, and should be appropriately addressed in the revision. I maintain that both the theoretical contributions and practical applications deem the paper worthy of being accepted, and will hence maintain my score.

**Claims And Evidence:**

The claims and evidence appear convincing to me.

**Essential References Not Discussed:**

Please consider discussing and citing "Binary Search with Distributional Predictions" by Dinitz et al. (https://arxiv.org/abs/2411.16030) which appeared at NeurIPS 2024. That work seems quite relevant with the topic of this paper.

**Experimental Designs Or Analyses:**

I did not find any glaring discrepancies in the experiments. They appear sound to me.

**Methods And Evaluation Criteria:**

To the best of my knowledge, yes.

**Other Comments Or Suggestions:**

1) Correct me if I am wrong, but it seems like the guarantee in Theorem 2.8 of each node $x$ having expected depth $O(\log(m/f_x))$ holds for any worst-case sequence of keys (even if they are not drawn iid from a distribution). It might be worth mentioning this somewhere after the Theorem.

2) Maybe I missed it, but I don't think there is clear mention of the fact that the optimal static cost (as established by Mehlhorn 1975) is on the order of the Shannon entropy of the distribution. You mention in Theorem 2.8 that the cost of your treap is $O(\sum_x f_x \log(m/f_x))$ matches the optimal static BST cost, but you haven't really stated anywhere what the static optimal BST cost is (unless I missed it) for the reader to really understand why this cost this optimal.

**Other Strengths And Weaknesses:**

As mentioned above, the paper proposes a new way of building a randomized tree data structure which is optimal in the sense of building a static tree for accessing keys drawn from a distribution. This is a worthwhile contribution. More specifically, the paper strictly improves upon the past result of Lin et al. 2022---while their construction of the treap was competitive against the statically optimal tree only for the Zipfian distribution, the present construction is competitive against the statically optimal tree for all distributions. Moreover, the present construction has nice robustness properties (smooth degradation in terms of KL divergence), and also extends to B-trees. The study considered in the dynamic setting also appears novel, and could inspire similar future research. Finally, the experimental results do appear promising, and show the practical gains in using the proposed treap.

While the technical contributions seem undeniable, I do however feel that the paper could be written better. For example, I felt that the introduction was quite dense. In its present form, a lot of jargon is stated without having defined/described certain concepts. As a nitpicking example, it is not clear to me why the abstract itself has a precise form of the priority function (Note that this is assuming the reader is familiar with the notion of a priority function). My general complaint is that a lot of technical sentences are written without easing the reader into what the words in the sentence mean/without setting up background/motivation. I would suggest going through the exercise of placing yourself in the shoes of an uninitiated reader, and going through the paper--this might suggest necessary modifications. Another piece of feedback is that the section on dynamic treaps is very hard to read technically, and doesn't really offer the reader with much insight. I would really encourage the authors to consider writing this section in a more accessible manner.

**Questions For Authors:**

1) Could you please comment on the robustness properties of the treap constructed in Lin et al. 2022? I think the paper warrants a discussion/comment about this, either in the section where you do the robustness analysis of your treap, or at any other place where you cite Lin et al. 2022

2) I am likely failing to see something simple, but could you elaborate on why your $O(\log(1/w_x))$ bound on the depth from Theorem 2.4 translates to an $O(\log n)$ bound? This would mean that $w_x \ge 1/n$---why is this true? You mention this in lines 309-310 in the section where you analyze alternate priority functions.

**Relation To Broader Scientific Literature:**

The study of data structures is fundamental in computer science, and at its core, the paper proposes a new way of building a randomized tree data structure which is optimal in the sense of building a static tree for accessing keys drawn from a distribution. This is a worthwhile contribution. The work also has relevance in the context of building data structures in a manner inspired by machine-learned predictions, which is also highly relevant in present contexts.

**Theoretical Claims:**

I only glanced over the proofs, and did not verify calculations line-by-line. They appear correct to me, and the progression in the overall analysis checks out to me. In particular, I could not really understand the calculations in the section on the dynamic setting, because this section is quite technically involved, and not really written in an accessible way.

---

> ### Author Rebuttal · Authors · 2025-03-30
>
> Thank you for your comments! We sincerely apologize for the various writing issues and will make every effort to improve them, particularly in the sections on the dynamic setting and the introduction. We will also cite the article you recommended and clearly state the static optimal BST cost to help readers better understand the context. Your writing suggestions are very helpful, and we truly appreciate you pointing them out. We address your questions as follows.
>
>
> 1. The robustness in [Lin et al.] is defined as follows. If the frequency predictor satisfies $\frac{1}{\delta}f_i\leq \hat{f_i} \leq \delta f_i$ for some constant $\delta>0$, then the corresponding Treap with predicted score has an additive constant error compared to the optimal.
>
>     If we use the same definition, our data structure also achieves an additive constant error in access cost relative to the optimal. However, our notion of robustness provides a **stronger guarantee** -  even when the predicted frequency does not satisfy this assumption. In particular, we still obtain an upper bound on the access cost measured by the KL divergence.
>
>
>
> 2. In Line 310, our intention was to illustrate that there exists a distribution under which the BSTs in Lin et al. have an $\Omega(n)$ access cost, whereas our data structure achieves $O(\log n)$. We apologize for the unclear wording, which may have led to the misunderstanding that our method always guarantees $O(\log n)$ worst-case access cost.
>
> We will revise Line 310 as follows to clarify this point:
>
> >However, it does not generally hold - there exists a distribution $p$ where the expected access cost for Lin et al. is $\Omega(n)$, while our data structure achieves only $O(\log n)$ cost.

---

### Official Review · Reviewer_AYCV · 2025-03-14

**Overall Recommendation:** 4

**Summary:**

The paper uses learned predictions to enhance the performance of classical data structures. Specifically, it presents learning-augmented binary search trees (BSTs) and B-Trees that leverage predictions about item access frequencies. Building on the work of Lin et al. [ICML’22], which proposed augmenting treaps (BSTs where nodes are arranged by key and randomized priorities) and B-treaps (their non-binary generalization) by replacing random priorities with predicted frequencies, this paper introduces a composite priority scheme. In particular, it assigns priorities as a function of predicted frequencies combined with random noise. This modification allows the structure to achieve static optimality more broadly, beyond the special cases addressed by Lin et al.
In the second part, the authors extend their approach to design a data structure that achieves the working-set bound, thus capturing temporal locality in access patterns. The paper also establishes robustness guarantees, showing that the performance of the proposed structures degrades gracefully under prediction errors. Finally, experimental results are presented to support the theoretical findings and to highlight the practical benefits of the proposed data structures over classical baselines.

**Claims And Evidence:**

Yes, the main claims of the paper are supported by detailed proofs and are presented in a clear and convincing manner. The authors carefully prove the static optimality bounds, robustness guarantees, and working-set bound, each supported. The use of KL divergence to quantify the impact of prediction error is particularly interesting and aligns with standard measures in learning-augmented algorithm literature.

In addition, the paper includes experimental results that support the theoretical findings and demonstrate empirical improvements over both classical and previously proposed learning-augmented baselines. Although the experimental evaluation is somewhat limited in scope and focused on synthetic data, it still provides consistent evidence in support of the proposed methods.

Overall, the paper’s claims are well-supported, logically coherent, and grounded in both theory and empirical validation.

**Essential References Not Discussed:**

When discussing optimal static trees, the classic work of Knuth [Acta Informatica, 1971] may be cited. Additionally, two recent works on learning-augmented skip lists, which also explore topics such as static optimality, are Zeynali et al. [ICML 2024] and Fu et al. [ICLR 2025] (only the second one is mentioned briefly in the context of experiments).

**Experimental Designs Or Analyses:**

Yes, I reviewed the experimental results presented in the paper. The experiments are based on synthetic access patterns generated from standard distributions. The performance metric used (average search depth) is well-aligned with the theoretical objectives studied in the paper. One limitation is that no explanation, discussion, or analysis of the experimental results is included in the main paper; this material is deferred to the appendix. As a result, there is a little disconnect between the theoretical development and the empirical evaluation, which weakens the overall presentation.

**Methods And Evaluation Criteria:**

Yes, the proposed methods and evaluation criteria are appropriate for the problem. The paper focuses on enhancing classical data structures using predictions, and the evaluation is based on standard theoretical metrics such as expected depth, working-set bound, and robustness with respect to prediction error (measured via KL divergence), all of which are meaningful in the context of search trees.

The experimental evaluation, while limited to synthetic datasets, is consistent with the theoretical goals of the paper. The selected access distributions are standard for illustrating both static and dynamic behaviors of the proposed data structures. However, the inclusion of more diverse or real-world access patterns could improve the empirical component.

**Other Comments Or Suggestions:**

- Line 024, the BST of Melhorn does not work with "estimates" of key frequencies; it assumes exact frequencies provided.
- I did not identify any typos or presentation issues; the paper is generally well-written and clean in terms of language quality and spelling.

**Other Strengths And Weaknesses:**

The paper presents a conceptually clean and technically solid contribution, combining classical data structure theory with modern learning-augmented algorithmic techniques. The integration of prediction-driven priority schemes into search trees and B-trees is interesting and improves the prior work of Lin et al. [ICML'22] in a meaningful way. That said, the empirical component is weakly integrated with the rest of the paper. A more cohesive and thorough treatment of the experimental results would strengthen the paper, especially for an ICML audience. This is particularly relevant given the practical importance of B-trees in real-world systems, including database indexing and file system implementations.

**Questions For Authors:**

1- The learning-augmented data structures presented in this paper achieve static optimality and the working-set property. However, these properties have already been established for classical data structures that do not rely on predictions, most notably splay trees. Could you clarify whether your data structures offer any notable theoretical advantages over splay trees?

2- What is the maximum depth of a node in your data structure? Is it possible that it gets linear depth under a highly skewed frequency distribution? In Line 096, you criticize the structure proposed by Lin et al. for having "super-logarithmic depth" in some cases, and this seems to be established in Theorem 2.14, where you demonstrate that assigning predicted frequencies directly as priorities (as in Lin et al.) results in linear depth and cost. How does this result compare to Proposition 2.2 in Zeynali et al. [ICML 2024]?

3- The structure proposed by Lin et al. [ICML 2022] appears to be deterministic, whereas your approach is randomized. Is there any known lower bound or negative result suggesting that deterministic data structures cannot achieve static optimality under prediction error?

**Relation To Broader Scientific Literature:**

The paper contributes to the growing literature on learning-augmented data structures, where algorithmic performance is improved through predictions. It builds directly on the recent work of Lin et al. [ICML 2022], who introduced the idea of using predicted access frequencies to assign priorities in treaps. This paper advances this line of work by introducing a composite priority scheme (combining predictions and random noise), which achieves static optimality more broadly across arbitrary frequency distributions.

The paper also situates itself within the broader field of self-adjusting and adaptive data structures, connecting classical ideas such as treaps, working-set bounds, and temporal locality with prediction-based techniques. The use of KL divergence to capture robustness under prediction error is interesting and aligned with recent trends in learning-augmented algorithm analysis, where "smooth" degradation with respect to prediction quality is a central theme.

**Theoretical Claims:**

Yes, I reviewed the proofs, including the analysis supporting the static optimality bounds, the robustness guarantees based on KL divergence, and the working-set bound in the dynamic setting. The arguments appear sound and are based on standard techniques in the analysis of search trees and learning-augmented algorithms. The proofs are clearly written, and I have not found any issues.

---

> ### Author Rebuttal · Authors · 2025-03-30
>
> Thanks for your thoughtful comments! We would like to begin by sincerely thanking you for pointing out the related work that we had overlooked. We will make sure to include proper citations in the updated version of the paper. Below, we address the three specific questions you raised:
>
>
> 1. Splay trees are conjectured to be dynamically optimal, which requires no prior knowledge. Therefore, it is very likely that no BST with/without learning-augmented can beat them in the theoretical bound. However, it is important to note that the runtime guarantees of splay trees are amortized. In particular, accessing a single item $x$ may still take linear time, which could be prohibitive in many scenarios.
>
>     Furthermore, there are no "splay trees" in the B-tree area under the external-memory setting. This makes our approach more applicable in such settings.
>
>     Our experiment results also show that splay trees have a higher access cost across most datasets. In this way, our work contributes to the study and understanding of static and dynamic optimality in search trees and could serve as a useful step toward a better understanding of splay tree optimality.
>
>
> 2. When the frequency distribution is highly skewed, some items may have an exponentially small frequency. This can result in depths as large as $O(n)$. However, the data structure still achieves *static optimality*.
>
>     In contrast, for the data structure proposed by Lin et al., Theorem 2.14 shows a frequency distribution where the depth is $\Omega(n)$. This aligns with the consistency results of $\Omega(n/\log n)$ given in Proposition 2.2 of Zeyanali et al..
>
>
> 3. Exploring lower bounds for deterministic data structures in the learning-augmented setting is indeed an interesting and important direction. However, to the best of our knowledge, there are currently no established results in this area. We agree that this is a valuable question for future work.

---

> > ### Comment · Reviewer_AYCV · 2025-04-02
> >
> > Thank you for the clarifications. I appreciate the results presented in the paper and will keep my current score.

---

### Official Review · Reviewer_HPYf · 2025-03-17

**Overall Recommendation:** 3

**Summary:**

Authors propose the following kind of binary search trees:
receiving prediction about the frequencies of all items,
their algorithm produces a tree which achieves static optimality
(up to a constant factor) and its cost smoothly deteriorates
with prediction error.
They extend their result to B-trees.
Authors also propose predictions of working set size to produce
a dynamic B-tree, where $B \geq \omega(\log n)$, which satisfies
working-set bound and its cost smoothly deteriorates with prediction
error.

**Claims And Evidence:**

Their claims are rigorously proved. However, have some troubles
understanding the claims about the robustness of their data structures.
I wonder whether there might be a misunderstanding on my side and authors
can explain that.
Looks to me that the statements of their theorem do not imply that
their tree is never worse (even with very bad predictions) than a balanced
tree built without any information about the input sequence.
E.g., they seem to claim that their tree has $O(\log n)$ worst-case depth (line 310),
but I do not see how does Thm 2.13 claim this.

**Essential References Not Discussed:**

I am not aware of any omissions.

**Experimental Designs Or Analyses:**

Authors replicate experiments from the previous theoretical work. I find
this enough for a mainly theoretical paper.

**Methods And Evaluation Criteria:**

methods and evaluation criteria look reasonable.

**Other Comments Or Suggestions:**

A suggestion:
you seem to call robustness what other papers in the area call
smoothness or dependence on prediction error. Might be better
to use the same terminology as other works.

**Other Strengths And Weaknesses:**

Strengths:

* authors study an important problem which is not yet understood in the
learning-augmented setting.

* authors remove a restrictive assumption of previous works which
required the input sequence coming from a zipfian distribution.

* apart from static optimality, the authors make a step towards dynamic
setting, evaluating the performance of their data structure in terms of the
working-set bound.

Weaknesses:

* at this moment, it is not clear to me whether their static BST
is robust, i.e., does not pay much more than m*log(n) regardless of the
quality of the predictions.

* their dynamic data structure requires a large branching factor
  (omega(log n)), i.e., it is far from a binary tree

**Questions For Authors:**

Can you please explain the behavior of your data structures (both static and
dynamic) with very bad predictions?

Looking at Theorem 2.13, if the real frequency of some element was a constant
(0.1, let's say) and the predicted frequency was exp(-n),
then the bound of Thm 2.13 on the access cost of your data structure
would be m*n while any balanced tree pays at most m*log(n). I.e., with very
bad predictions, your tree seem to be much worse than a balanced tree built
without any information about the input sequence, and is not "robust",
in the sense commonly used e.g. in the survey of Mitzenmacher and
Vassilvitskii which you cite in your paper.

**Relation To Broader Scientific Literature:**

Topic of this paper are relevant and fits well within existing
literature on learning-augmented algorithms.
In particular, BST is a prominent problem and deserves study in
the learning-augmented setting.

**Theoretical Claims:**

I did not check any proof completely, but the arguments look reasonable.

---

> ### Author Rebuttal · Authors · 2025-03-30
>
> Thanks for your comments! We will address your concern as follows.
>
> **Robustness and Bad Predictions**
>
> We use the word *robustness* to describe how the performance of our data structure degrades smoothly with respect to the prediction error - measured by KL divergence in the static setting and mean absolute error in the dynamic setting.
>
> Using the robustness definition in [1, 2] - i.e., the competitive ratio under the worst-case prediction - our data structure is $O(n)$-robust in the static setting (Proposition 2.3 in [2]).
>
> However, the data structure achieves $O(\log n)$-robustness if we ensure that each predicted score is at least $1/n^C$ for a large constant $C>0$. In practice, this can be achieved by modifying the predictor to guarantee it, for example, by taking a maximum of the predicted score and $1/n$.
>
> In the dynamic setting, assuming the predicted scores satisfy $\tilde{w}_{i,j}\geq 1/\text{poly}(n)$ as in Theorem 4.5, the data structure achieves $O(\log n)$-robustness.
>
>
> **Worst-case Bound**
> In Line 310, our intention was to illustrate that there exists a distribution under which the BSTs in [3] have an $\Omega(n)$ access cost, whereas our data structure achieves $O(\log n)$. We apologize for the unclear wording, which may have led to the misunderstanding that our method always guarantees $O(\log n)$ worst-case access cost.
>
> We will revise Line 310 as follows to clarify this point:
>
> >However, it does not generally hold - there exists a distribution $p$ where the expected access cost for [3] is $\Omega(n)$, while our data structure achieves only $O(\log n)$ cost.
>
>
> **Branching Factor**
> The branching factor arises from B-treaps. If we want to construct a BST in the dynamic setting, the analysis proceeds in the same way, without involving the branching factor.
>
>
> [1] Mitzenmacher, Michael, and Sergei Vassilvitskii. "Algorithms with predictions." Communications of the ACM 65.7 (2022): 33-35.
>
> [2] Zeynali, Ali, Shahin Kamali, and Mohammad Hajiesmaili. "Robust Learning-Augmented Dictionaries." International Conference on Machine Learning. PMLR, 2024.
>
> [3] Lin, Honghao, Tian Luo, and David Woodruff. "Learning augmented binary search trees." International Conference on Machine Learning. PMLR, 2022.

---

> > ### Comment · Reviewer_HPYf · 2025-04-02
> >
> > Thank you for your answers, in particular your clarification about the robustness of your algorithm. It is unfortunate that your algorithm is not robust in the sense that its worst-case competitive ratio O(n) is achieved by literally any algorithm and any static balanced tree achieves much better robustness of O(log n). I consider this the weakest point of your paper. I also see that the paper is well appreciated by other reviewers and I have decided to maintain my score.

---

### Decision · Program_Chairs · 2025-05-01

**Decision:**

Accept (poster)

**Comment:**

The paper shows that the performance of binary search trees and specifically treaps can be improved by using predictions about the frequencies of all items, achieving static optimality up to a constant factor. Though reviewers observed that the statements do not provide the usual guarantees of robustness in learning-augmented algorithms, KL divergence provides a meaningful quantification of the quality of advice and thus reviewers agree that the contributions of the paper are valuable.